# Genetic diversity of *Ralstonia solanacearum* causing vascular bacterial wilt under different agro-climatic regions of West Bengal, India

**Ankit Kumar Ghorai**[1⊙], **Subrata Dutta**[1⊙], **Ashis Roy Barman**[2⊙]*

**1** Department of Plant Pathology, Bidhan Chandra Krishi Viswavidyalaya, Mohanpur, Nadia, West Bengal, India, **2** Department of Plant Pathology, RRS (CSZ), Bidhan Chandra Krishi Viswavidyalaya, Akshaynagar, Kakdwip, South 24-Parganas, India

⊙ These authors contributed equally to this work.
* roybarman.ashis@bckv.edu.in

**Data Availability Statement:** All relevant data are within the manuscript and its Supporting Information files.

**Funding:** P.I. of the Project: Ashis Roy Barman Grant Number: File No. ECR/2017/000547, dated

## Abstract

The bacterial wilt disease of solanaceous crops incited by *Ralstonia solanacearum* is a menace to the production of solanaceous vegetables all over the world. Among the agro climatic zones of West Bengal, India growing solanaceous vegetables, the maximum and minimum incidence of bacterial wilt was observed in Red and Lateritic zone (42.4%) and Coastal and Saline zone (26.9%), respectively. The present investigation reports the occurrence of bacterial wilt of Bottle gourd by *R. solanacearum* Sequevar 1–48 for the first time in India. Two new biovars (6 and 3b) along with biovar 3 have been found to be prevalent in West Bengal. Under West Bengal condition, the most predominant Sequevar was I-48 (75%) followed by I-47 (25%). Low genetic variation (18.9%) among agro climatic zones (ACZs) compared to high genetic variation (81.1%) within revealed occurrence of gene flow among these ACZs. Standard genetic diversity indices based on the concatenated sequences of the seven genes revealed ACZ-6 as highly diverse among five agro climatic zones. The multi locus sequence analysis illustrated occurrence of synonymous or purifying selection in the selected genes in West Bengal and across world. Under West Bengal conditions maximum nucleotide diversity was observed for the gene *gyr*B. Occurrence of significant recombination was confirmed by pairwise homoplasy test (θ = 0.47*) among the RSSC isolates of West Bengal, belonging to Phylotype I. Phylotype I isolates of West Bengal are involved in exchange of genetic material with Phylotype II isolates. In case of worldwide RSSC collection, eleven significant recombination events were observed among the five phylotypes. Phylotype IV was genetically most diverse among all the Phylotypes. The most recombinogenic phylotype was Phylotype III. Further, the most diverse gene contributing to the evolution of RSSC worldwide was observed to be endoglucanase (*egl*).

## Introduction

*Ralstonia solanacearum* (Smith 1896) Yabuuchi et al. 1995 is a destructive phytopathogenic bacterium that causes vascular wilt in several economically important crops and ornamentals.

2610.2017 Funding Authority: Science and Engineering Research Board (SERB), Department of Science and Technology, Government of India URL: http://www.serb.gov.in/home.php The funders had no role in study design, data collection and analysis, decision to publish, or preparation of the manuscript.

**Competing interests:** The authors have declared that no competing interests exist.

This vascular bacterial wilt was first described by E. F. Smith in potato, tomato and brinjal and further classified the bacterium under the genus *Bacillus* as *Bacillus solanacearum* in 1896 [1]. But with the advent of taxonomic positioning the bacterium was transferred to the class β-proteobacteria [2] with its new name *Ralstonia solanacearum*. Initially *R. solanacearum* was classified into races on the basis of its ability to attack various hosts [3]. Race classification was followed by grouping of the strains on the basis of biochemical properties. Hayward (1964) has described four biotypes of *R. solanacearum* on the basis of differential ability of acid production and denitrification [4]. Currently, the strains of *R. solanacearum* are defined to be a heterogeneous species or a species complex [5]. Species complex is defined as strains of a species which "are the product of long evolution occurring independently in various areas on different hosts" [6]. As a species complex, termed as *Ralstonia solanacearum* species complex (RSSC), it is distinguished into four monophyletic clusters called phylotypes on the basis of ITS sequences [5]. Phylotype I, equivalent to the Division I [7], includes Asian strains of race 1 (biovars 3, 4, and 5). Phylotype II is equivalent to division II [7] and includes race 1, 2, and 3 from America (biovars 1, 2 and N2). Phylotype III includes strains belonging to biovars 1 and 2T isolated from Africa, Reunion Island and the Madagascar Islands in the Indian Ocean. Phylotype IV is constituted by strains belonging to biovars 1, 2 and 2T from Indonesia, Australia and Japan. Further, the strains of the same phylotype cluster together on the basis of partial endoglucanase gene similarity to form sequevars [5]. Till date 57 sequevars have been reported for RSSC [8]. Huge economic losses incurred worldwide and absence of effective management practices against this bacterium has finally led it to be designated as A2:58 (high risk) phytopathogen by the European and Mediterranean Plant Protection Organization (EPPO), in order to regulate and restrict its spread across the globe [9].

Presently scanty or no clear picture is available over the present status of diversity of RSSC strains among the different agro climatic zones of West Bengal. Efforts have not been undertaken till date to investigate the contribution of various genes behind the evolution of the bacterium and possibility of genetic recombination that may occur among the strains of RSSC under West Bengal condition. The distribution of biovars and search for possible prevalence of new biovars in West Bengal, India, needs much attention. Again, survey for finding out new possible hosts and sequevars have not yet been undertaken in West Bengal during the past years. Taking all these into consideration, our present investigation focuses to decipher the population structure, genetic diversity and contribution of genes towards the evolution of RSSC among the five major solanaceous crop growing agro climatic zones of West Bengal.

## Materials and methods

### Survey and collection of wilted plants

Isolation of *Ralstonia solanacearum* species complex (RSSC) was done from 5 agro-climatic zones (ACZ) covering 17 districts of West Bengal, where bacterial wilt is historically a major menace. Isolation was done from diverse host *viz.*, brinjal, chilli, capsicum, tomato, marigold and bottle gourd. Isolation was done from plants with foliar epinasty or wilt symptoms after confirmation with ooze test. Top 2–3 cm portion of the stem pieces were collected using disinfected secateurs and kept in zip lock bags inside ice cooled boxes for transport to laboratory. The stem pieces were washed in sterile distilled water (SDW) followed by surface sterilization with 0.1% $HgCl_2$. After drying in blotting paper, ooze was collected in SDW inside 25 ml conical flasks for 30 minutes. Loop full of ooze was streaked in zigzag manner over Casamino acid-Peptone-Glucose–Triphenyl tetrazolium chloride media (CPG-TZC) (casamino acid, 1 g $L^{-1}$; peptone, 10 g $L^{-1}$; glucose 10 g $L^{-1}$ and agar 15 g $L^{-1}$ added with 0.005% v/v 2,3,5-triphenyl tetrazolium chloride 50 μg $mL^{-1}$) [10]. Incubation was provided for 48 hours at 28°C. Single

colony of *R. solanacearum* showing virulent, fluidal, irregular and creamy white with pinkish centre were further streaked on CPG-TZC media for purification. The slimy virulent cultures were maintained in 30% glycerol stock at—80°C for future use. Further, single colonies of each isolates were suspended in 1 ml sterile water and maintained at room temperature [11].

## Designation of races

Tobacco seeds were grown for 25 days in seedling trays containing autoclaved sterile cocopeat and then transplanted to 6-inch pots filled with autoclaved soil. Tobacco leaves aged 30 days were infiltrated with needleless hypodermal syringe by gently pressing the barrel mouth against bottom of lower leaf surfaces [12]. The leaf infiltration was done in 10 places of the interveinal area of lamina. *R. solanacearum* strains were freshly grown on CPG-TZC media [10] by incubating at 28 ±1°C for 2 days. Inoculum was prepared by suspending loop full of virulent slimy exopolysaccharide secreting colonies in nuclease free double distilled water (ddH$_2$O). The cell density of the bacterial suspension was adjusted to Optical Density (OD) of 0.01 (approximately $10^7$–$10^8$ Cfu ml$^{-1}$) by measuring OD at 600 nm. Infiltration was done on three leaves of a test plant for each of the 36 strains of *R. solanacearum*, isolated from various hosts. Observations were taken 12, 24, 36, 48 and 192 hours post leaf infiltration and race determination was performed according to the scheme following Lozano and Sequirea, 1970 [12].

## Biovar characterisation

Improved biovar test was conducted using phenol red as pH indicator [13]. Working basal media of 100 ml was prepared by adding 1ml of the hundred times stock solutions of NH$_4$H$_2$PO$_4$ (10%), KCl (2%), MgSO$_4$·7H$_2$O (2%), Difco Bacto peptone (10%), phenol red (0.8% by adding one drop of 5 N NaOH in order to dissolve the dye and maintain neutral pH of 7.4) and 0.3 g of agar into 95ml of ddH$_2$O.The basal media was autoclaved and cooled to 65°C. Meanwhile, 20% filter sterilised solution of sugars (lactose, maltose and cellobiose) or sugar alcohols (mannitol, sorbitol and dulcitol) were prepared. This was followed by mixing 19 parts of cooled autoclaved basal media with 1 part of 20% filter sterilised solution of carbo-hydrate solutions, in order to get a final concentration of 1% carbohydrate. 150 μl of the mixed media was then poured into the wells of autoclaved micro titre plates using sterilised 8 channel micropipette and aluminium foil boat. Inoculum was prepared by streaking each of 36 *R. sola-nacearum* isolates on CPG-TZC media [10]. The freshly grown strains were suspended in nuclease free ddH$_2$O and cell density adjusted to OD$_{600}$ of 0.01 (approximately $10^7$ – $10^8$ Cfu ml$^{-1}$). Twenty micro litres of the *R. solanacearum* suspension was used to inoculate the mixed test media, followed by covering of the micro titre plates with sterile cover and incubation at 32°C. The experiment was conducted twice with three replications for each strain of collected *R. solanacearum*. Observations were taken after every 24 hours till colour changes were noticed (Table 1).

## Pathogenicity test

Inoculum was prepared from isolates preserved in room temperature. All the 36 *R. solana-cearum* isolates were freshly grown in CPG-TZC media [10] at 28°C for 48 hours. Characteristic single slimy colonies of each isolate showing exopolysaccharide secretions were re-cultured on CPG media without TZC for 48 hours at 28 ±1$^0$ C. The cells were plated by serial dilution technique in CPG-TZC media [10] and inoculum load of $10^8$cfu/ml was prepared. Pathogenicity test was performed using host *viz*., brinjal (cv. Muktokeshi), tomato (cv. Pusa Ruby) and Chilli (cv. Pusa Jwala). Seedlings were prepared in green house and transplanted after 3 weeks

Table 1. **Biovar determination of *R.* *solanacearum*** isolates based on the acid production through utilization of various carbohydrates.

| Substrate | Biovars | | | | | | |
|---|---|---|---|---|---|---|---|
| | 1 | 2 | 3 | 4 | 5 | 6 | 3b |
| Cellobiose | - | + | + | - | + | + | + |
| Maltose | - | + | + | - | + | + | + |
| Lactose | - | + | + | - | + | + | - |
| Dulcitol | - | - | + | + | - | - | - |
| Sorbitol | - | - | + | + | - | + | + |
| Mannitol | - | - | + | + | + | + | + |

+: Utilisation of substrate; -: Unutilised substrate

into pots containing sterilised soil. The lateral roots of the seedlings were given injury prior transplanting. The prepared inoculum of strength $10^8$cfu/ml was poured surrounding the root zone by the method of soil drenching [14]. Five seedlings of each host plants were inoculated for each isolate in different pots. Control was maintained by soil drenching with sterilised double distilled water. The inoculated seedlings were watered regularly and maintained in plant growth chamber at 28–30˚C with 12 hours light and dark cycle and 85% relative humidity. Development of symptom was monitored after every week for 1 month.

## DNA isolation

Single colony of each isolate of *R. solanacearum* was inoculated into 20 ml of CPG broth in 50 ml conical flasks. The cultures were incubated overnight at 28±1˚C with constant shaking at 200 rpm. The bacterial cells were harvested in the form of pellet in the micro centrifuge tubes after centrifugation for 10 min at 9167 g. The pellets were provided a single wash with 1 M NaCl followed by three washes with sterile double distilled water. Proteinase K (50 μg/ml) of 200 ml volume was prepared for DNA extraction using 10 mM Tris-HCL (pH 7.5) and 1 M EDTA (pH 8.0). Bacterial pellets were dissolved in 400 μl Proteinase K solution by gentle taping followed by incubation at 56˚C for 15 min in hot water bath. Deactivation of enzyme activity was done at 80˚C for 15 min. The mixture was transferred instantly on ice and kept for 5 min followed by centrifugation at 15493g for 5 min. The supernatant containing the DNA was collected in a separate sterile micro centrifuge tube, properly tagged and preserved at −20˚C for future use. Quantification of the purified DNA was done by loading 2μl of purified DNA onto the lower optic surface of the spectrophotometer (Thermo Scientific NanoDrop™ 1000). The quality of the purified DNA was checked by measuring the ratio of absorbance *i.e.*, A260/A280 and also by gel electrophoresis in 0.8% agarose gel stained with ethidium bromide.

## Molecular confirmation

**16S rDNA sequencing and phylogenetic analysis.** The purified DNA was subjected to amplification of the 16S rRNA gene of approximately 1500 bp with the forward primer 27f and reverse primer 1525r [15]. The 25 μl reaction mixture consisted of 1 μl template DNA, 2.5 μl 10X PCR buffer with preadded $MgCl_2$ (Takara), 0.5μl dNTPs (2.5 mM each, Takara), 1 μl of each primer (10 mM), 0.3 μl Takara Taq DNA polymerase (5 U μl$^{-1}$) and rest 18.7 μl of molecular grade nuclease free $ddH_2O$. The PCR conditions in the thermal recycler Quanta Biotech (QB-96) is mentioned in the S1 Table. The PCR product (4 μl) were loaded with 6X orange loading dye (Thermo Scientific) and allowed to run in 1% agarose gel under constant voltage of 80V for 45 minutes. Molecular ladder of 100bp (Himedia MBT130-200LN) was used to

determine the molecular size of the amplicon produced. All sequences of 16SrRNA region were submitted to NCBI GenBank and accession numbers were obtained.

The purified DNA were then subjected to species confirmation test by conducting amplification with RSSC specific primers OLI 1 and Y2 [16] complementary to conserved partial 288 bp region of 16S rDNA. PCR reaction mixture was prepared as mentioned above and the thermal conditions maintained are mentioned in S1 Table. The PCR products were visualised by electrophoresis in 1.5% agarose gel, prepared in 1X TAE buffer.

## Phylotyping

Phylotypes of the RSSC isolates collected from five agro-climatic zones of West Bengal were determined by conducting phylotype specific multiplex PCR (Pmx-PCR) using a set of phylotype specific oligo nucleotide primers based on 16S-23S ITS region [5] (Table 2). PCR master mixture for reaction was prepared as mentioned earlier. The primer mixture consisted 4 pmol each of OLI1 and Y2; 6 pmol each of Nmult 21:1F, Nmult 21:2F and Nmult 22-InF; 10 pmol of Nmult 22-RR and 18 pmol of Nmult23-AF. The following PCR conditions were maintained: an initial denaturation step at 96˚C for 5 min, 35 cycles containing denaturation at 94˚C for 15 sec, annealing at 59˚C for 30 sec, extension at 72˚C for 30 sec, final extension step at 72˚C for 10 min and holding at 4˚C.Gel electrophoresis was done by running 4 μl of Pmx-PCR product with 6X orange loading dye in 1.5% agarose gel. A constant electric field of 80 V with 1 hour of run time was provided for gel electrophoresis. Ladder of 100bp (Himedia MBT130-200LN) was used to characterise the molecular size of the amplicon generated specific to a phylotype.

## Sequevar determination

For determination of sequevars, the isolated DNA was subjected to amplification of 850 bp of the partial endoglucanase gene located on the 2.7 Mb mega-plasmid. In addition to this, the 1417 bp partial portion of *hrp*B gene (regulatory transcription regulator) was amplified using the primer pair RShrpBf and RShrpBr [17]. The master mix was prepared as mentioned earlier and the PCR conditions along with primer information are mentioned in S1 Table. Designation of sequevar to the RSSC isolates of West Bengal were done by constructing phylogenetic tree with already reported sequevar sequences of *egl* and *hrp*B gene of worldwide RSSC strains.

## Genetic variability analysis on the basis MLSA of RSSC

Total seven genes were selected for Multi Locus Sequence Analysis (MLSA). These genes included 4 chromosomal housekeeping genes (1. *gyr*B: DNA gyrase subunit B, 2. *adk*: adenylate kinase, 3. *gdh*A: glutamate dehydrogenase oxidoreductase, 4. *leu*S: leucine tRNA ligase and 5. *pps*A: phosphoenol pyruvate synthase) and two megaplasmid virulence related genes (6. *egl*: endoglucanase precursor and 7. *fli*C: encoding flagellin protein). Information about the primer

**Table 2. List of primers and amplicon size for Pmx-PCR used in phylotyping of RSSC.**

| Primers | Sequence (5'-3') | Specific to phylotype | Amplicon |
|---|---|---|---|
| Nmult21:1F | CGTTGATGAGGCGCGCAATTT | Forward primer for phylotype I | 144 bp |
| Nmult21:2F | AAGTTATGGACGGTGGAAGTC | Forward primer for phylotype II | 372 bp |
| Nmult23:AF | ATTACSAGAGCAATCGAAAGATT | Forward primer for phylotype III | 91 bp |
| Nmult22:InF | ATTGCCAAGACGAGAGAAGTA | Forward primer for phylotype IV | 213 bp |
| Numult22:RR | TCGCTTGACCCTATAACGAGTA | Reverse primer for all phylotypes | |
| OLI1 | GGGGGTAGCTTGCTACCTGCC | Species Specific forward primer | 288 bp |
| Y2 | CCCACTGCTGCCTCCCGTAGGAGT | Species Specific reverse primer | |

pairs used to amplify these seven genes has been mentioned (S1 Table). Amplification of all the genes were done by preparing reaction mixture as mentioned earlier. The PCR conditions followed for the amplification of these seven genes are mentioned in the S1 Table. The amplicons were examined by electrophoresis through ethidium bromide stained agarose gels in TAE buffer for each gene. Molecular ladder of 100bp (Himedia MBT130-200LN) was used to determine the molecular size of the amplicon produced.

The amplified products for all the genes were outsourced to Agri Genome Labs Pvt Ltd., Kochi, for enzymatically purification and partial Sanger dideoxy sequencing using Big Dye Terminator v3.1 Cycle Sequencing Kit on ABI 3730xl Genetic Analyzer. The sequences retrieved were subjected to similarity search using the Basic Local Alignment Search (BLASTn) tool administered in NCBI. All sequences of these genes were submitted to NCBI GenBank through the tool Bankit and the accession numbers were obtained.

## Sequence analysis

Alignment and trimming were performed with ClustalW [18] tool of MEGA v7 [19]. Concatenation of FASTA formatted, aligned and trimmed multiple gene sequences were performed using various text editing features of BioEdit (v7.0.5.3) [20] and MEGA 10. Phylogenetic trees were constructed using both Neighbour joining method (NJ) and Maximum Likelihood (ML) method using Jukes-Cantor algorithm administered in MEGA 7. Bootstrapping was conducted with 1000 iterations for the NJ tree and ML tree.

Our investigation included gene sequences of 112 strains. This consisted 36 strains (C1) isolated from five agro-climatic zones of West Bengal and 76 reference strains (C2) belonging to all five phylotypes of RSSC, retrieved from the nucleotide GenBank database of NCBI. Analysis of the genetic diversity was done in three levels; agroclimatic zone (ACZ), genes and phylotype. For this, the standard parameters of genetic diversity *viz.* number of polymorphic sites (s), % percentage of polymorphism (p), average number of nucleotide diversity (k), nucleotide diversity ($\pi$), number of haplotypes (h) and haplotype diversity (Hd) were estimated using the DNAsp v6.12 [21].

Analysis of Molecular Variance was conducted using Arlequin 3.5 [22] in order to explore contribution of among and within population variations. Further, fixation index and pairwise Fst based on Nei's genetic distance were also computed using Arlequin 3.5, in order to illustrate genetic diversity and differences among populations.

In order to investigate the evolutionary forces acting over each gene under study, the neutrality tests like Tajima's D, Fu and Li's D were performed. The method of estimating the $K_a$/$K_s$ ratios and $d_n$/$d_s$ ratios [23] using the MEGA v7 were further carried out to understand the type of substitution [19].

Presence of recombination in the phylotypes and individual genes along with concatenated gene set was investigated by performing the pairwise homoplasy index test ($\emptyset_w$) [24] administered in Splitstree4 [25]. Neighbour net tree was constructed using Splitstree4 programme to investigate reticulations.

The RDP package [26] was used to identify recombination events occurring among genes between strains of different phylotypes. Eight type of programmes administered in RDP4 *viz.*, RDP, GENECOV, Bootscan, Maxchi, Chimaera, SiSscan, PhylPro, LARD and 3seq were used to identify recombination points. The settings of all these programmes were kept as default and a Bonferroni corrected P value cut-off of 0.05 was considered to detect optimal recombination events.

The LDHAT module [27] administered in RDP4 [26] was used to compute the rate of mutations and recombinations of different genes in phylotypes. The initial value of $\rho$ was

determined by conducting a first run with ρ value of 30. The obtained initial value of ρ was then utilised for successive runs. Block penalty values were fixed to 5, 10, 20, 30 and 40. The cut-off for the frequency of Minor allele was set to 0.05.

Percolation networks were constructed using SIDIER [28] programme of R package which can interpret indels and substitution matrix and finally express evolutionary events as percolation network.

## Results

### Survey and isolation of the pathogen

The survey was conducted from the year of 2017 to 2018 and bacterial wilt samples were collected from various hosts like brinjal, chilli, tomato, capsicum, bottle gourd and marigold. Among the 17 districts surveyed, the per cent disease incidence (PDI) ranged from 7.5% (Uttar Dinajpur) to 53.3% (Murshidabad) with average of 38.42% on the year 2018 and 7.25% (Uttar Dinajpur) to 45% (Malda) with average of 30.1% on the year 2017 (Fig 1). For both the years, the mean PDI was higher in Malda, Murshidabad, Coochbehar, Burdwan, Bankura, Jhargram, Purulia and Nadia. Among the five agro climatic zones, the PDI ranged from 27.45% (ACZ-5, Coastal and saline zone) to 42.4% (ACZ-6, Red and lateritic zone) in 2018 and 20.09% (ACZ-5) to 32.6% (ACZ-6) in 2017 (Fig 2). Isolation of the RSSC was done from infected samples of various hosts like brinjal, chilli, tomato, capsicum, bottle gourd and marigold collected from the survey over various districts of West Bengal. A total 36 pure cultures of *R. solanacearum* were isolated from 5 agro climatic zones (ACZ) covering 17 districts of West Bengal. Among them, 4 isolates were from ACZ-2, 14 from ACZ-3, 9 from ACZ-4, 3 from ACZ-5 and remaining 6 isolates were from ACZ-6 (Table 3).

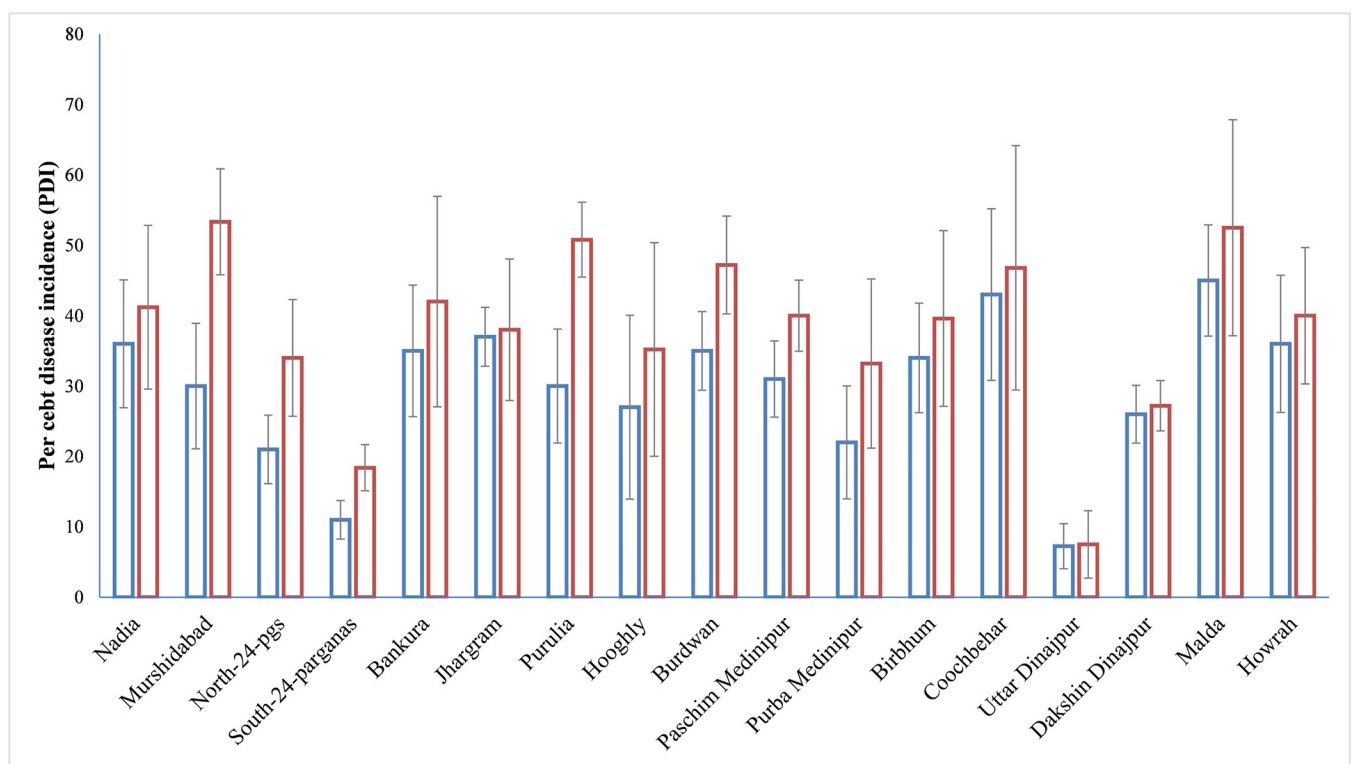

**Fig 1. Prevalence of vascular bacterial wilt (VBW) in different districts of West Bengal.** Bars with blue and red boundary represent prevalence of VBW in 2017 and 2018, respectively.

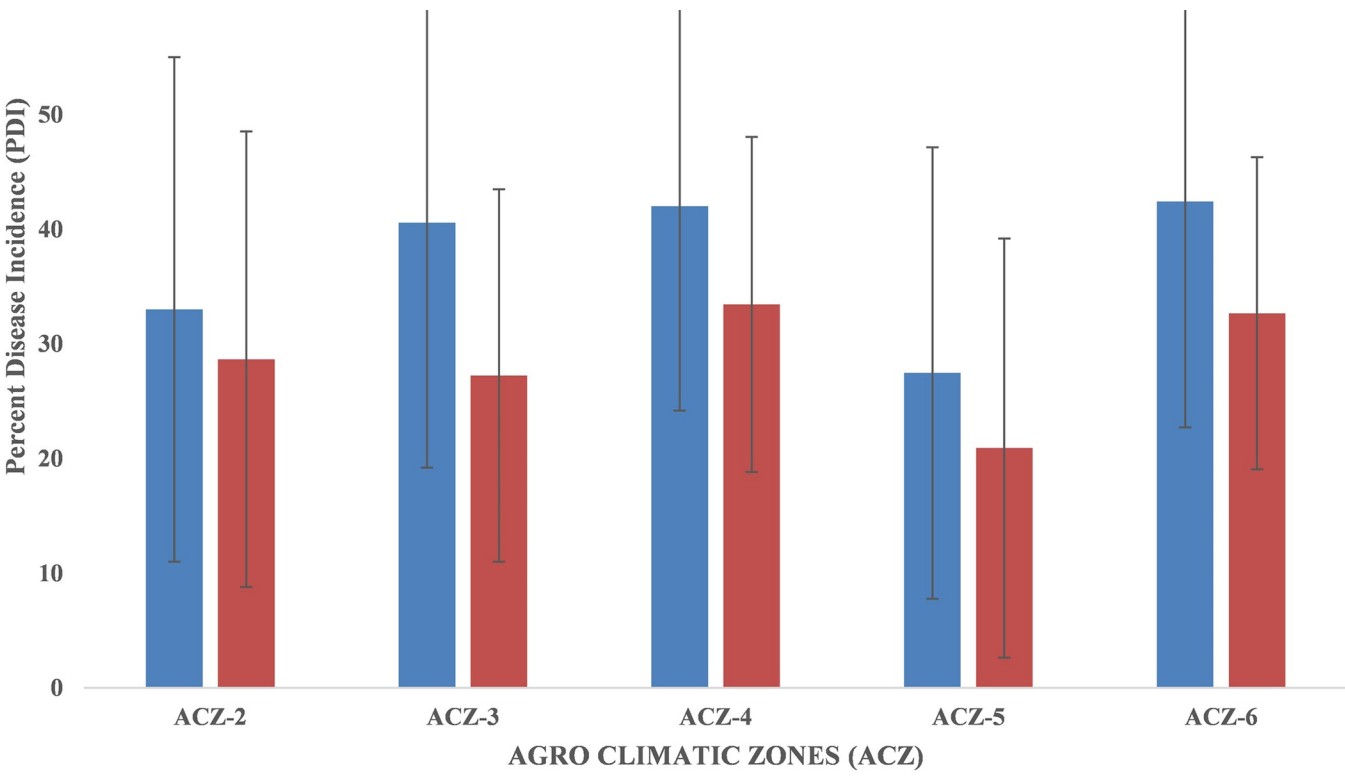

**Fig 2. Prevalence of VBW in different agro climatic zones (ACZ) of West Bengal.** Bars with red and blue colours represent prevalence of VBW in 2017 and 2018, respectively.

### Designation of races

Thirty days old healthy tobacco seedlings were infiltrated with prepared inoculum of RSSC isolates collected from various agro climatic zones of West Bengal. Dark brown lesions surrounded by yellow halo started developing 48 hours after leaf infiltration. Necrotic symptoms were observed after 60 hours. Finally, wilting of the whole plant was observed after 7–8 days (Fig 3). All these phenomena were in line with the inoculation experiment on tobacco with race 1 as reported earlier [12]. Hence, all the 36 isolates of RSSC collected from West Bengal were categorised to Race 1.

### Biovar characterisation

Utilisation of carbohydrates was illustrated with change of red coloured phenol red based basal media (pH:7.4) to yellow coloured resultant acidic by-products (pH:6.8) with reduction in pH of the culture media. The change in colour of the culture media in the wells of micro-titre plates permanently stopped 3 days post inoculation. Among the 36 isolates, 18 isolates (50%) were categorised as biovar 3b, which were unable to utilise dulcitol and lactose. Nine isolates were categorised as biovar 6 which were unable to utilise dulcitol. The remaining 9 isolates were categorised as biovar 3 which were able to utilise all the sugars and alcohols (Table 3; Fig 4). The biovar 3 and 3b were prevalent in all the five ACZs, while the biovar 6 was not found in ACZ-2 and ACZ-5.

### Pathogenicity test

Pathogenicity test was conducted for the 36 isolates of RSSC isolated from seven different hosts, which were collected from five agro climatic zones of West Bengal. Inoculation was

**Table 3. Information about race, biovar, sequevar, phylotype and pathogenicity group of RSSC isolates of various agroclimatic zones (ACZ) of West Bengal.**

| Isolate | Host | ACZ | District, block and village | R[a] | Bv[b] | Phy[c] | Seq[d] | PG[e] |
|---------|------|-----|-----------------------------|------|-------|--------|--------|-------|
| T1 | Tomato | 3 | Nadia, Chadamari, Muratipur | 1 | 6 | 1 | I48 | I |
| B2 | Brinjal | 3 | Nadia, Chakdaha, Simurali | 1 | 7 | 1 | I48 | I |
| B5 | Brinjal | 3 | Nadia, Ranaghat 1, Aistala | 1 | 6 | 1 | I48 | I |
| B8 | Brinjal | 3 | Nadia, Chakdah, Madanpur | 1 | 3 | 1 | I48 | I |
| B9 | Brinjal | 3 | Nadia, Santipur, Phulia | 1 | 6 | 1 | I48 | I |
| B10 | Brinjal | 3 | Nadia, Krishnanagar, Mayakol | 1 | 6 | 1 | I48 | II |
| T11 | Tomato | 3 | Murshidabad, Beldanga-I, Mokrampur | 1 | 3 | 1 | I48 | II |
| B12 | Brinjal | 3 | Murshidabad, Murshidabad Jiaganj, Bahadurpur | 1 | 3 | 1 | I48 | II |
| B13 | Brinjal | 3 | Murshidabad, Berhampore, Sasidharpur | 1 | 6 | 1 | I48 | I |
| T14 | Tomato | 3 | Murshidabad, Beldanga-II, Rezinagar | 1 | 3b | 1 | I48 | II |
| B15 | Brinjal | 3 | Murshidabad, Sagardihi, Baburgram | 1 | 3b | 1 | I48 | III |
| B16 | Brinjal | 4 | Murshidabad, Khargram, Uttar Gopinathpur | 1 | 3b | 1 | I48 | I |
| B17 | Brinjal | 5 | Purba Medinipur, Kontai-1, Badhia | 1 | 3 | 1 | I48 | I |
| T18 | Tomato | 5 | Howrah, Bagnan-1, Sitalpur | 1 | 3 | 1 | I48 | II |
| B19 | Brinjal | 4 | Birbhum, Mayureshwar-II, Kotasur | 1 | 6 | 1 | I48 | II |
| C20 | Chilli | 4 | Hooghly, Pandua, Digha | 1 | 6 | 1 | I47 | IV |
| T21 | Tomato | 4 | Malda, Habibpur, Chak Dahari | 1 | 6 | 1 | I48 | I |
| Bo22 | Bottlegourd | 4 | Hooghly, Dhaniakhali, Khajurdaha | 1 | 3b | 1 | I48 | II |
| Ca23 | Capsicum | 4 | Hooghly, Polbadatpur, Amra | 1 | 3b | 1 | I48 | I |
| T24 | Tomato | 4 | Dakshin Dinajpur, Gangarampur, Jahangirpur | 1 | 3b | 1 | I47 | II |
| B26 | Brinjal | 3 | North-24-Parganas, Amdanga | 1 | 3b | 1 | I48 | II |
| B27 | Brinjal | 3 | North-24-Parganas, Bongao, Polta | 1 | 3b | 1 | I48 | II |
| B28 | Brinjal | 6 | Bankura, Bankura 1, Kenjakura | 1 | 3b | 1 | I48 | I |
| B30 | Brinjal | 6 | Purulia, Kashipur, Gagnabadh | 1 | 3 | 1 | I48 | I |
| B32 | Brinjal | 4 | Bankura, Indus, Palasdanga | 1 | 3 | 1 | I48 | II |
| B34 | Brinjal | 6 | Jhargram, Gopiballabhpur-I, Allampur | 1 | 3b | 1 | I47 | II |
| M36 | Marigold | 3 | Nadia, Ranaghat-II, Huda | 1 | 3b | 1 | I47 | II |
| B39 | Brinjal | 4 | Burdwan, Memari-1, Amadpur | 1 | 3b | 1 | I48 | I |
| B41 | Brinjal | 2 | Coochbehar, Tufanganj-1, Chilakhana | 1 | 3b | 1 | I48 | I |
| B42 | Brinjal | 5 | South-24 Parganas, Kakdwip, BCKV, RSS | 1 | 3b | 1 | I47 | I |
| B43 | Brinjal | 2 | Jalpaiguri, Falakata, Falakata | 1 | 3 | 1 | I48 | I |
| B44 | Brinjal | 2 | Coochbehar-I, Petbhata, Magfala | 1 | 3b | 1 | I47 | II |
| B45 | Brinjal | 2 | UttarDinajpur, Itahar, Kamrdanga | 1 | 3 | 1 | I48 | III |
| B50 | Brinjal | 6 | Paschim Medinipur, Garbeta-1, Phulberia | 1 | 6 | 1 | I47 | I |
| C51 | Chilli | 6 | Burdwan, Ausgram-II, Ramnagar | 1 | 3b | 1 | I47 | I |
| C52 | Chilli | 6 | Burdwan, Kanksa, Panagarh | 1 | 3b | 1 | I47 | IV |

[a] Race

[b] Bioavar

[c] Phylotype

[d] Sequevar

[e] Pathogenicity group

done using soil drench method of inoculation with inoculum load of $10^8$cfu/ml [14]. Symptoms like drooping down of top leaves were observed and multiple leaves were found to droop in successive weeks (Fig 3). Among the 36 isolates, 50% (18 isolates) were classified under the pathogenic group I which were capable to incite wilting symptoms in all the three test hosts *viz.*, brinjal, tomato and chilli. Fourteen isolates (38.9%) were able to infect only brinjal and

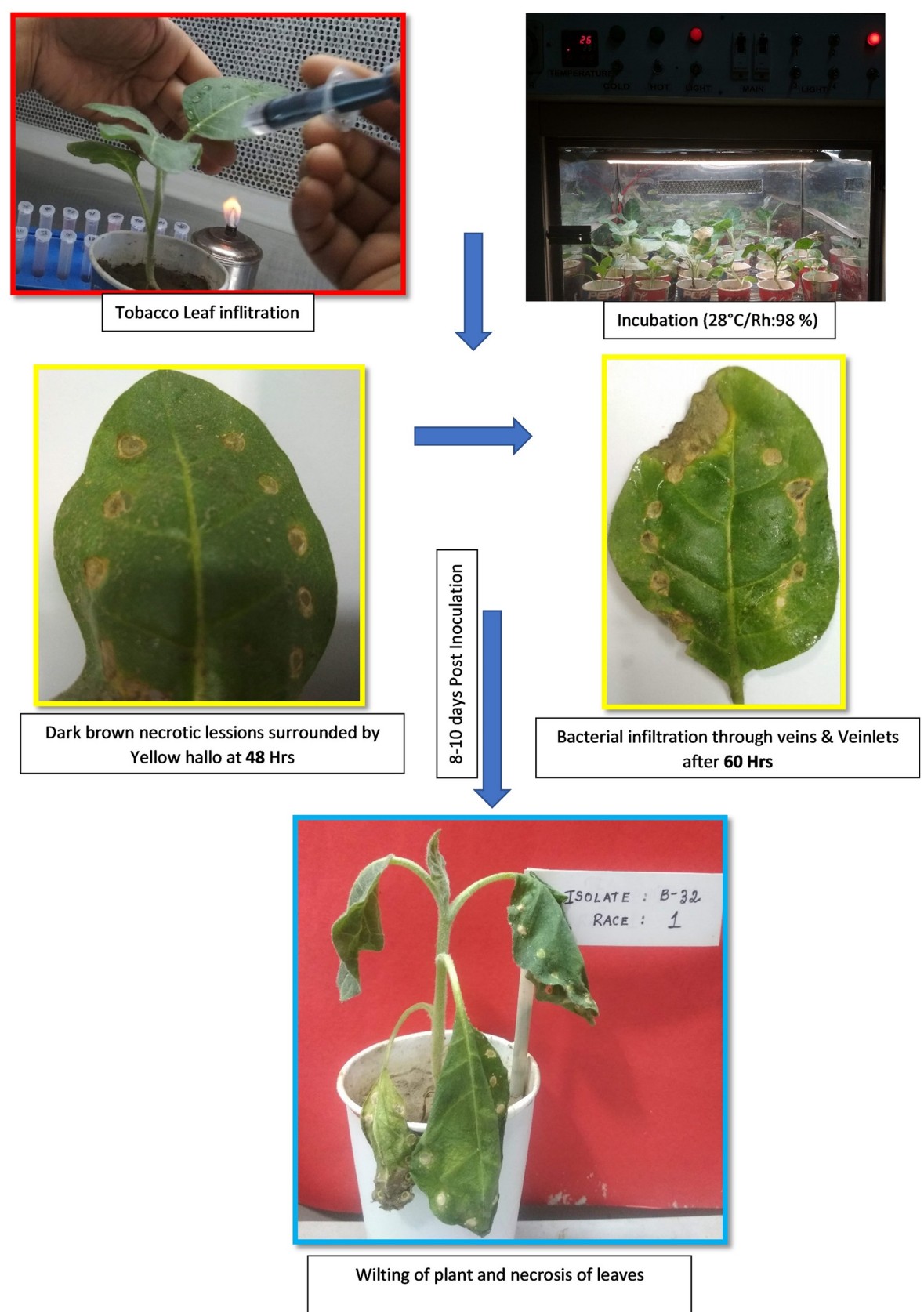

**Fig 3. Different phases of symptom development in tobacco used for race determination.**

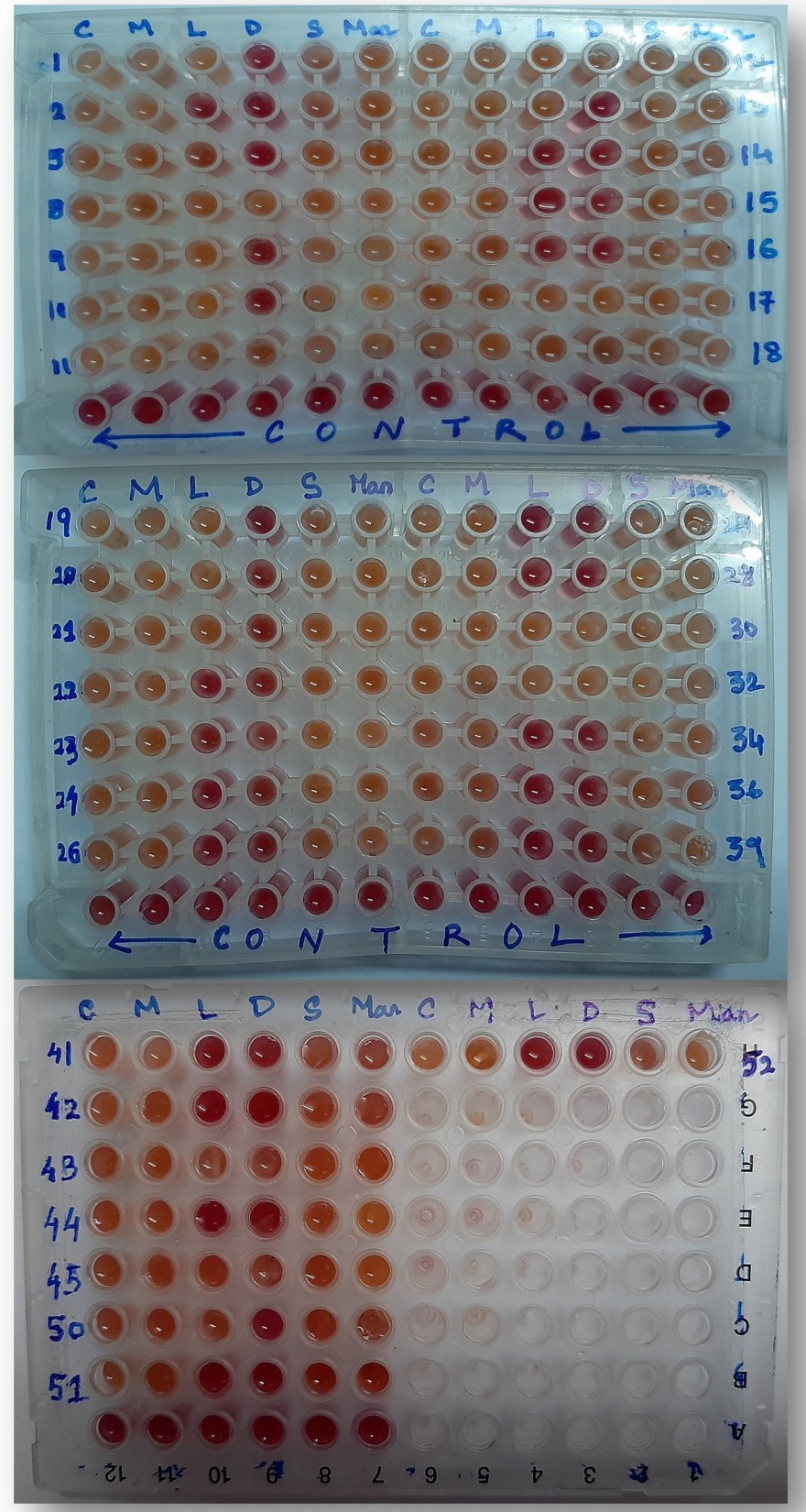

**Fig 4. Improved biovar test showing colour changes on sugar and alcohol utilisation.**

tomato test plants and thus classified into the pathogenic group II. Two isolates from brinjal (B15 and B45) were capable of infecting only brinjal test plants and were grouped under the pathogenic group III and the remaining two isolates from chilli (C20 and C52) were found to incite wilt symptoms only on chilli and were grouped under the pathogenic group IV (Table 3). All the test plants showing symptoms of wilting were revalidated by performing ooze test.

Isolates of brinjal and tomato were found to produce wilting symptoms as early as 7 days post inoculation (DPI). The isolates of chilli took a period of minimum 15 DPI to induce wilting of chilli plants. The isolate of capsicum induced wilting of chilli 7 DPI. The isolate of *R. solanacearum* from bottlegourd (Bo22) was grouped into pathogenic group II and initiated wilting on brinjal at 7 DPI and tomato at 10 DPI. The isolate of marigold (M36) was also classified under the pathogenic group II (Table 3).

## Confirmation of RSSC

The purified DNA of the isolates were amplified with the *Ralstonia solanacearum* specific primers OLI1-Y2 [16]. All the 36 isolates were confirmed as *Ralstonia solanacearum* based on the amplification of 288 bp fragment (S1 Fig).

To further confirm the isolates to be *R. solanacearum* and to identify the sub-group to which they belong, the 16S rDNA regions were amplified (S2 Fig) and Sanger sequencing was performed. BLASTn tool of NCBI identified all the 36 isolates of West Bengal as *R. solanacearum*, having 98 to 100 per cent similarity with 16s rDNA sequences of *R. solanacaerum* present in GenBank databases. The sequences of the isolates were submitted to NCBI and Accession numbers were obtained (S2 Table).

The phylogenetic tree (Fig 5) constructed with MEGA7.0 grouped all the 36 isolates of West Bengal into division 1 with its representative Australian isolate (ACH0171) belonging to biovar 3, Indonesian isolate (R791) belonging to biovar 3 and Japanese isolate (MAFF211266) belonging to biovar 4. Again, this division 1 is also referred to as Phylotype 1 in the new hierarchical system based on partial sequencing of *egl*, *hrp*B and *mut*S genes [5]. As 16SrRNA is highly conserved region with less amount of polymorphism (3.07% in our study) we further characterised the isolates on the basis of partial *egl* and *hrp*B gene sequencing.

## Phylotyping by multiplex PCR

The phylotype specific multiplex PCR [5] produced amplicons of approximately 144 bp and 288 bp specific to Phylotype I (Asiatic origin) in case of all the 36 isolates of *R. solanacearum* (Table 3). Gel documentation of six representative isolates belonging to five agro climatic zones have been represented in S3 Fig. Hence, irrespective of different host of isolation, all the isolates of West Bengal were found to belong to Phylotype I. These findings are in line with 16SrDNA sequencing which showed that all the isolates belong to Division-I, corresponding to Phylotype I of RSSC [5].

## Sequevar determination

The endoglucanase gene of 850 bp was amplified for all the 36 isolates (S4 Fig) and partial sequencing of the amplified product was performed. Based on the NCBI-BLAST results, the sequences were confirmed as *egl* of *R. solanacearum*. Sequences were submitted to GenBank through Bankit tool and Accession numbers were obtained (S2 Table). Evolutionary history of *R. solanacearum* was inferred utilising 655 nucleotides of the endoglucanase gene (*egl*) sequences (77% of whole *egl* gene) of 36 isolates of West Bengal and 90 reference sequences retrieved from NCBI database belonging to 51 sequevars identified previously [29]. The

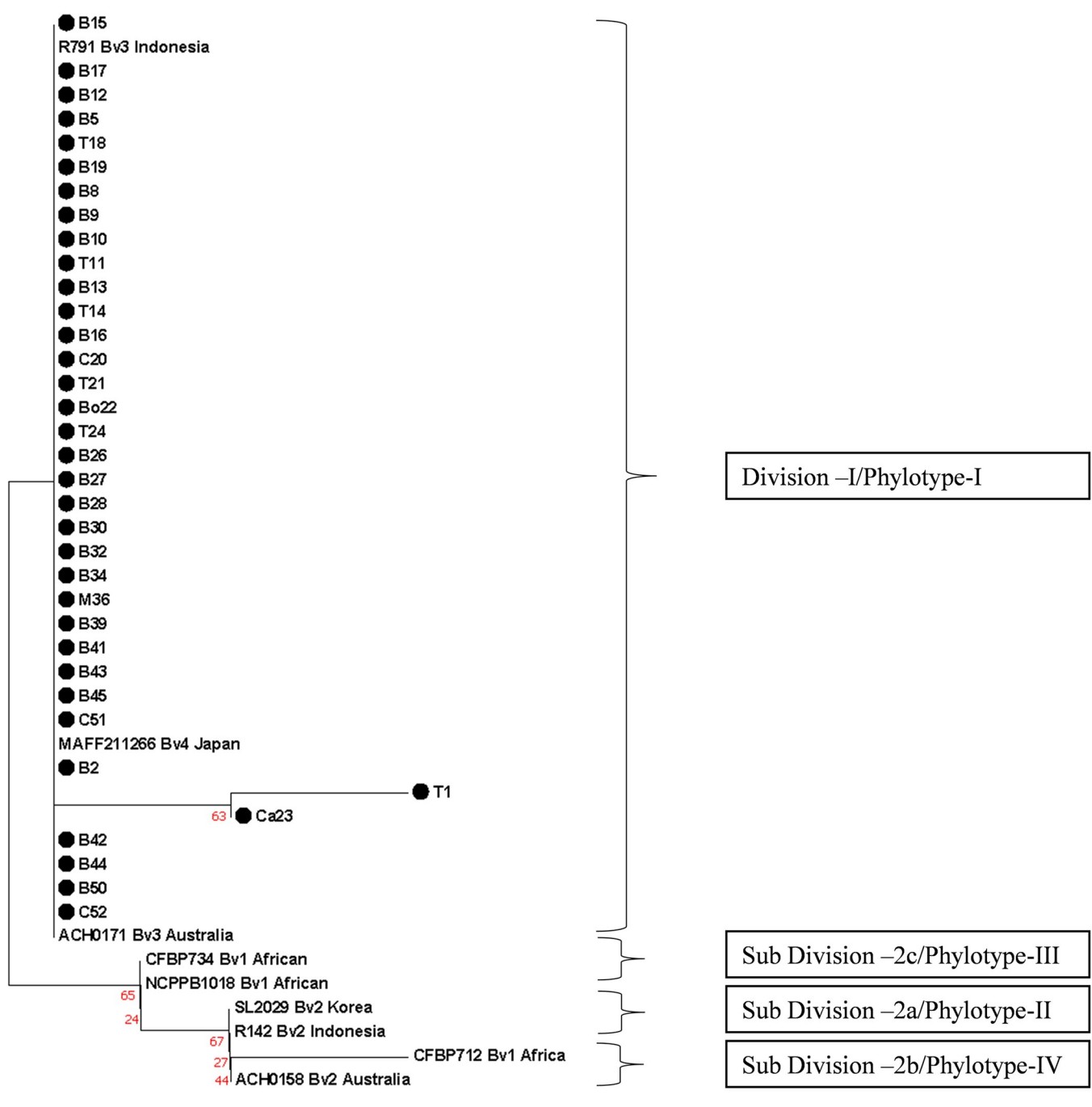

**Fig 5. Dendogram constructed using partial 16S rDNA sequences showing phylogenetic relationships of 36 isolates of RSSC from West Bengal with reference sequences from various phylotypes of *R. solanacearum*.** The phylogenetic tree was constructed using Neighbour joining method using MEGA7 with sum of branch length = 0.34. Bootstrap values showing percentage of replicate trees with which the associated taxa clustered together are shown next to the branches.

reference sequences were considered to allocate isolates under this study to sequevars. The evolutionary tree categorised all the 36 isolates into the clade Phylotype I (Asian strains), which was consistent with the results of Pmx-PCR and 16SrDNA sequence analysis. Both the NJ and ML trees produced similar branching patterns and hence only NJ tree has been

presented (Fig 6). Branch separating the clade of Phylotype I from closest Phylotype IV have a boot strap value of 99%. More over the split-up bootstrap value for Phylotype I was 95%, which shows that the tree is highly robust. Among the 36 isolates, 27 isolates grouped with reference strain M2 which was previously defined as Sequevar I-48. The rest 9 isolates grouped with the reference strain GMI8254 which was previously defined as sequevar I-47 (Table 3). The two groups containing sequevar I-47 and I-48 were separated from each other with high bootstrap value (53%). Two isolates B28 and B30 have formed a separate group with insignificant boot strap value of 8 per cent and also belonged to the same cluster with strain M2 (Accession No.: FJ561067). Hence these two isolates were finally characterised as sequevar I-48. Phylotype I was closely related to Phylotype IV and distantly related to Phylotype III and Phylotype II. The percentage of polymorphic sites in the *egl* gene sequences was 28.24% (185 polymorphic sites out of 655 nucleotide positions) as analysed by DNAsp 6.0.

## Partial sequencing of *hrp*B for sequevar identification

The 1417 bp region of the *hrp*B gene was amplified (S5 Fig) for all the 36 isolates collected from five agro climatic zone of West Bengal. Partial sequencing of the *hrp*B was done and all the 36 sequences showed 99–100 similarity with *hrp*B sequences of *R. solanacearum* present in NCBI database. All the sequences were submitted to GenBank through Bankit and Accession numbers were obtained (S2 Table). The evolutionary tree was generated using the 652 nucleotides of the *hrp*B gene sequences (which is 46% of whole *hrp*B gene) from 36 isolates of West Bengal and 12 reference sequences retrieved from NCBI database. The available reference sequences from NCBI nucleotide database were considered in the analysis with the sequences generated in the current experiments. The phylogenetic tree exhibited the clear separation of Phylotype I from Phylotype III with a high bootstrap value of 97% (Fig 7). The Phylotype I split up into the two sequevars I-47 and I-48 having boot strap value of 99%, which indicates that the tree is highly robust. Similar to the phylogenetic tree based on *egl* sequences (Fig 6), the 27 isolates were clustered with strain Rs-10-244 (JQ687197) which is previously defined as sequevar I-48. The remaining 9 isolates were clustered with the strain Rs-09-202 (JQ687189) which is defined previously as sequevar I-47. The percentage of polymorphic sites in the *hrp*B gene sequences was 17.9% (117 polymorphic sites among 652 nucleotide sites) as analysed by DNAsp 6.0. The overall information about characterisation of all 36 RSSC collected from various parts of West Bengal have been summarised in Table 3.

## Multi Locus Sequence Analysis (MLSA)

The genetic diversity of *R. solanacearum* isolates was analysed in three different levels, *i.e.*, among genes, agro-climatic zones (ACZ) and phylotypes. The analysis of genetic diversity was performed by conducting multi locus sequence analysis (MLSA) with the partial sequences of *adk* (381 bp), *egl* (654 bp), *fli*C (312 bp), *gdh*A (538 bp), *gyr*B (354 bp), *leu*S (699 bp) and *pps*A (624 bp). The length of the concatenated sequence was 3564 bp and altogether 112 sequences were divided into three groups, *viz.*, C1 (n = 36, sequences from the current experiment), C2 (n = 76, reference sequences retrieved from NCBI) and C1 and C2 combined together (C3). Accession numbers for all the sequences are enlisted in the S2 Table.

## Comparison of genetic diversity of RSSC between West Bengal and worldwide

All the genes under the collections C1 and C2 were polymorphic, except *leu*S in the RSSC collection of West Bengal C1 (n = 36). Among the isolates of West Bengal, the highest nucleotide diversity was observed for the gene *gyr*B ($\pi$ = 0.6%) followed by *adk* ($\pi$ = 0.3%) and *egl* ($\pi$ =

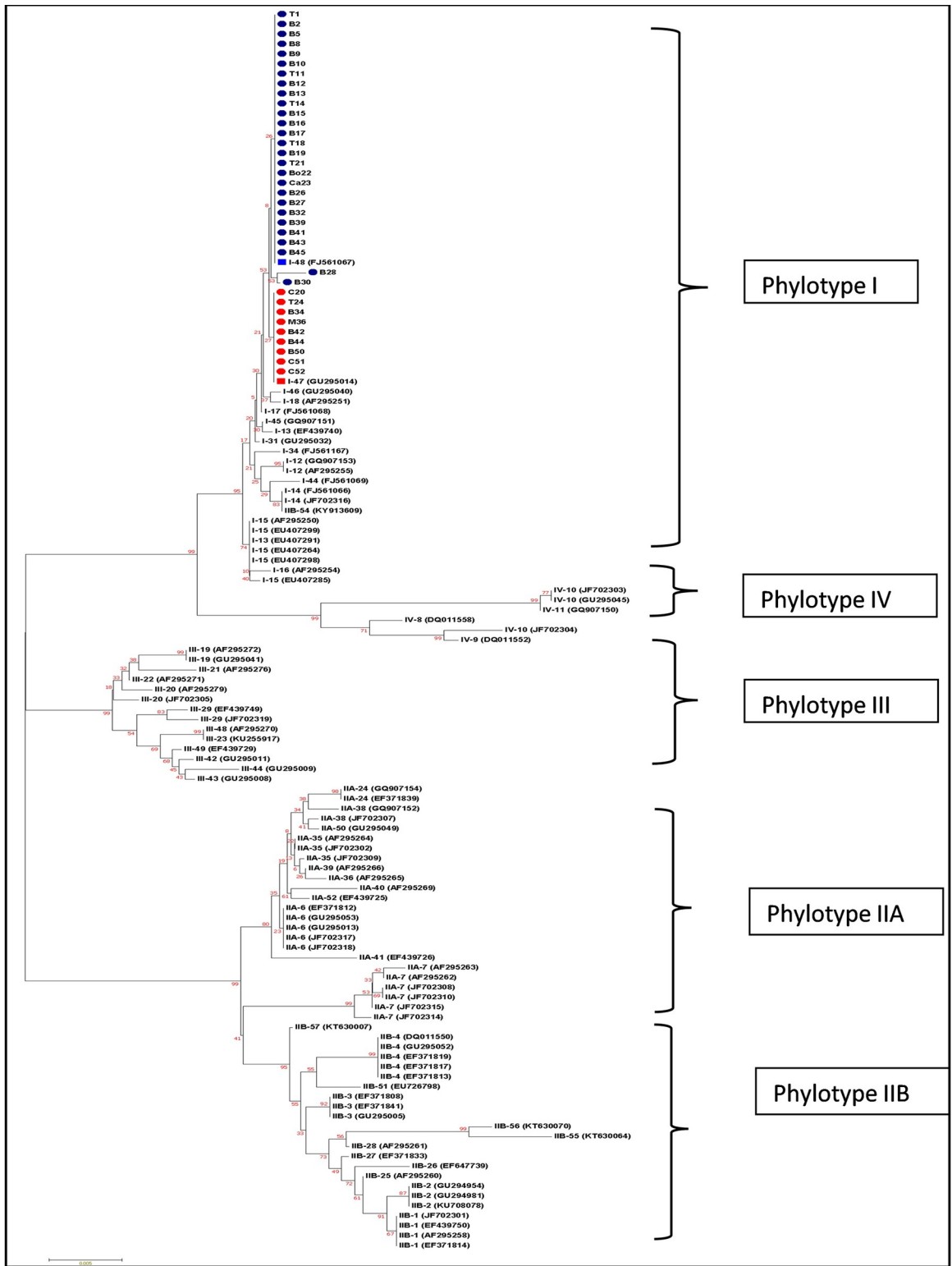

**Fig 6. Phylogenetic relationship of partial *egl* sequences (655 nucleotides) of 36 isolates of *R. solanacearum* from West Bengal and 90 reference strains belonging to all phylotypes obtained from NCBI databases.** Tree was constructed by MEGA 7 using Neighbour joining method and algorithm of Jukes and Cantor (1969). Numbers at branchpoints indicate percent bootstrap support for 1,000 iterations. The optimal tree with the sum of branch length = 0.20900382 is shown.

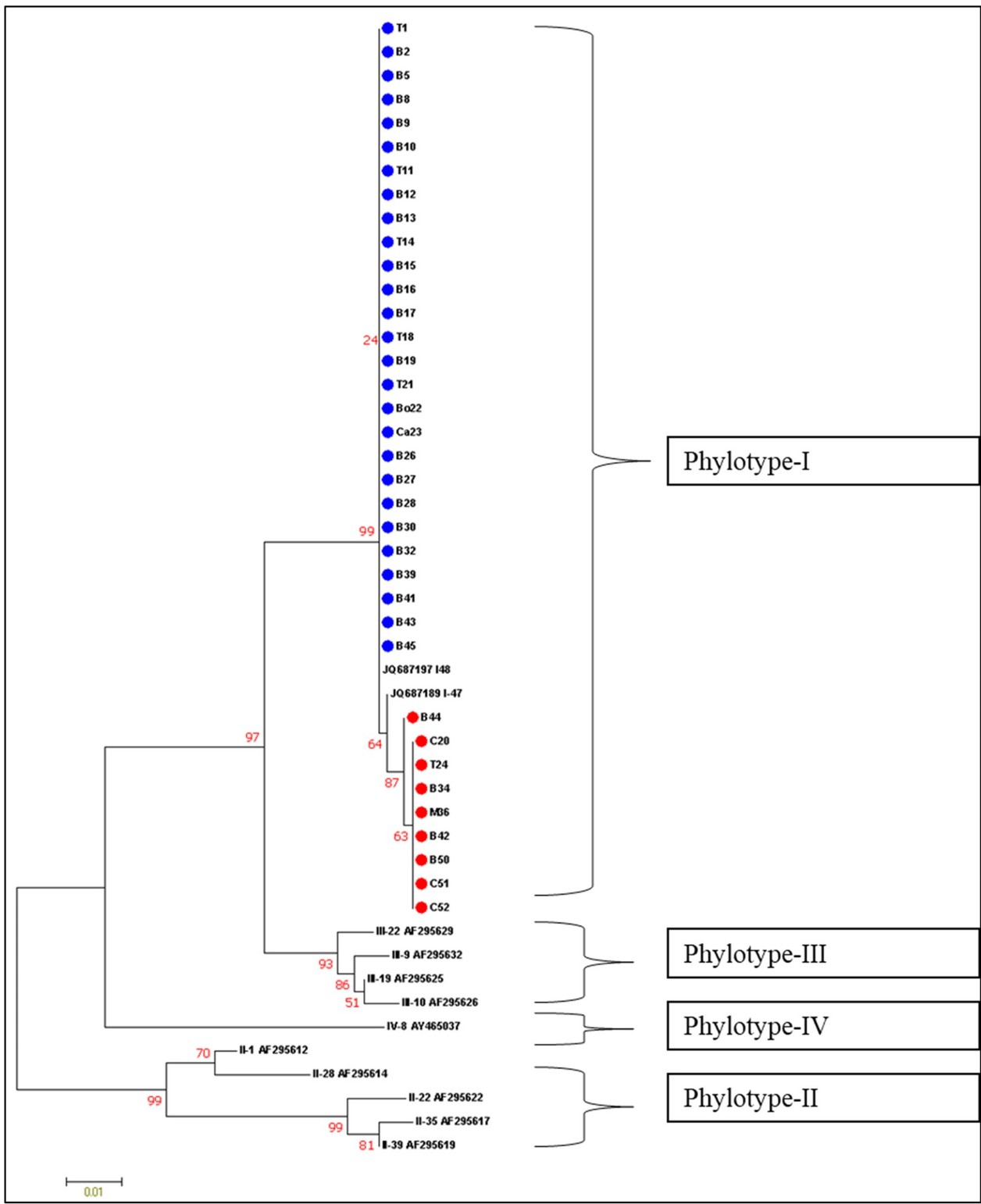

**Fig 7. Phylogenetic relationship of partial *hrp*B sequences (652 nucleotides) of 36 isolates of *R. solanacearum* from West Bengal and 12 reference strains belonging to all phylotypes obtained from NCBI databases.** Tree was constructed by Mega 7 using Neighbour joining method and algorithm of Jukes and Cantor (1969). Numbers at branchpoints indicate percent bootstrap support for 1,000 iterations. The optimal tree with the sum of branch length = 0.01 is shown.

**Table 4. Comparison of genetic properties of different loci used in MLSA for the RSSC isolates from West Bengal (C1), worldwide (C2) and C1 and C2 combined (C3).**

| Collection | Locus | TS[a] | NS[b] | S[c] | P[d] | K[e] | Π[f] | H[g] | Hd[h] | T[i] | $K_a$[j] | $K_s$[k] | $d_N$[l] | ds[m] |
|---|---|---|---|---|---|---|---|---|---|---|---|---|---|---|
| **C1 (n = 36)** | *Adk* | 36 | 381 | 4 | 1.05 | 0.98 | 0.003 | 4 | 0.42 | 0.04 | 0.003 | 0.002 | 1.18 | 2 |
| | *Egl* | 36 | 654 | 4 | 0.61 | 0.63 | 0.00097 | 4 | 0.47 | -0.84 | 0.0002 | 0.003 | 0.49 | 13.17 |
| | *fli*C | 36 | 312 | 2 | 0.64 | 0.11 | 0.00036 | 2 | 0.06 | -1.5 | 0 | 0.001 | 0.00 | 7.6 |
| | *gdh*A | 36 | 538 | 1 | 0.19 | 0.06 | 0.0001 | 2 | 0.06 | -1.13 | 0 | 0.0005 | 0.012 | 1 |
| | *gyr*B | 36 | 354 | 9 | 2.54 | 2.07 | 0.0058 | 4 | 0.48 | -0.14 | 0 | 0.02 | 0.00 | 17.66 |
| | *leu*S | 36 | 699 | 0 | 0.00 | 0 | 0 | 1 | 0 | 0 | 0 | 0 | 0.00 | 0 |
| | *pps*A | 36 | 624 | 6 | 0.96 | 0.39 | 0.0006 | 1 | 0.26 | -2.0* | 0.0002 | 0.002 | 0.89 | 12.9 |
| | *Conca-tenated* | 36 | 3564 | 26 | 0.73 | 3.84 | 0.0011 | 16 | 0.69 | -1.12 | 0.00004 | 0.005 | 0.91 | 75.6 |
| | | | | | | | | | | | $K_a/K_s$ | | $d_N/d_S$ | |
| **C2 (n = 76)** | *Adk* | 76 | 381 | 27 | 7.09 | 8.32 | 0.0218 | 43 | 0.97 | 0.87 | 0.12 | | 0.10 | |
| | *Egl* | 76 | 654 | 163 | 24.9 | 34.5 | 0.052 | 52 | 0.98 | -0.14 | 0.29 | | 0.15 | |
| | *fli*C | 76 | 312 | 34 | 10.9 | 6.86 | 0.0226 | 35 | 0.95 | -0.3 | 0.22 | | 0.10 | |
| | *gdh*A | 76 | 534 | 76 | 14.2 | 12.4 | 0.0239 | 30 | 0.94 | -0.81 | 0.15 | | 0.07 | |
| | *gyr*B | 76 | 354 | 36 | 10.2 | 7.98 | 0.0232 | 46 | 0.98 | -0.15 | 0.16 | | 0.06 | |
| | *leu*S | 76 | 699 | 81 | 11.6 | 15.6 | 0.023 | 37 | 0.96 | -0.41 | 0.11 | | 0.04 | |
| | *pps*A | 65 | 630 | 89 | 14.1 | 22.5 | 0.035 | 29 | 0.94 | 0.46 | 0.58 | | 0.03 | |
| | *Conca-tenated* | 76 | 3564 | 417 | 11.7 | 85.6 | 0.03 | 75 | 1 | -0.26 | 0.19 | | 0.09 | |
| **C3 (n = 112)** | *Adk* | 112 | 381 | 28 | 7.35 | 9.01 | 0.024 | 45 | 0.91 | 1.35 | 0.10 | | 0.11 | |
| | *Egl* | 112 | 654 | 165 | 25.2 | 31.3 | 0.047 | 55 | 0.94 | -0.23 | 0.30 | | 0.15 | |
| | *fli*C | 112 | 312 | 34 | 10.9 | 5.48 | 0.017 | 36 | 0.79 | -0.68 | 0.20 | | 0.10 | |
| | *gdh*A | 112 | 534 | 77 | 14.4 | 10.6 | 0.019 | 31 | 0.85 | -0.98 | 0.14 | | 0.07 | |
| | *gyr*B | 112 | 354 | 39 | 11.0 | 9.35 | 0.026 | 49 | 0.94 | 0.38 | 0.11 | | 0.05 | |
| | *leu*S | 112 | 699 | 81 | 11.6 | 13.9 | 0.019 | 37 | 0.87 | -0.5 | 0.11 | | 0.04 | |
| | *pps*A | 101 | 630 | 92 | 14.60 | 19.89 | 0.0329 | 34 | 0.88 | - | 0.06 | | 0.03 | |
| | *Conca-tenated* | 112 | 2934 | 424 | 14.45 | 1192.6 | 0.027 | 87 | 0.96 | -0.28 | 0.18 | | 0.09 | |

[a]Total sequences

[b] number of nucleotide sites

[c]number of polymorphic sites

[d] percentage of polymorphism

[e] average number of nucleotide differences

[f] nucleotide diversity

[g] number of haplotypes

[h] haplotype diversity

[i] Tajima's D

[j] non synonymous substitution

[k] synonymous substitution

[l] rate of substitutions at non-silent sites

[m] rate of substitutions at silent sites

*(p<0.05)

0.097%). Moderate level of polymorphism was obtained in *pps*A gene (π = 0.06%), *fli*C gene (π = 0.036%) and minimum in case of *gdh*A (π = 0.01%) and *leu*S gene (π = 0%) (Table 4). In the collection C2 comprising of sequences from worldwide isolates (n = 76), the highest nucleotide diversity was observed for the gene *egl* (π = 5%) as compared to other genes. The number of haplotypes, haplotype diversity and percentage of polymorphism were higher in the worldwide RSSC collection C2 (H = 75, Hd = 1 and p = 11.7) as compared to the RSSC collection of West

Bengal (C1) (H = 16, Hd = 0.69 and p = 0.73). Among the genes, the least diversity was observed for *leu*S (S = 0) in C1 and *adk* (S = 27) in C2. Considering both the collections (C1 and C2), the number of haplotypes, haplotype diversity and percentage of polymorphism were higher in C3 RSSC collection (C1 and C2 combined) (H = 87, Hd = 0.96 and p = 14.45) (Table 4).

Among the seven genes of RSSC collection of West Bengal (C1), the value for $K_a$ was observed for *adk*, *egl* and *pps*A whereas value for $d_N$ was obtained for *adk*, *egl*, *gdh*A, *pps*A and the concatenated sequence. Thus, non-synonymous substitutions (represented by $K_a$ and $d_N$) were not observed for all the genes under consideration. However, except *leu*S, synonymous substitutions (represented by $K_S$ and $d_S$) were observed for all the other genes and the concatenated sequence and the value of $d_N/d_S$ was found to be less than 1. The values of Tajima's D were negative for the five genes (*egl*, *fli*C, *gdh*A, *gyr*B and *pps*A), which revealed that the genes are under purifying selection. Although the Tajima's neutrality test showed positive value for the gene *adk* but it was found to be not significant. Again, no Tajima's D value (Tajima's D = 0) was obtained for the gene *leu*S. All these results inferred that the genes for RSSC isolates of West Bengal (C1) are under purifying selection.

While considering the worldwide collection of C2, all the values for $K_a/K_S$ and $d_N/d_S$ were found to be less than 1. Additionally, when we considered both the collections of RSSC of C1 and C2 together *i.e.*, C3 (n = 112), $d_N/d_S$ values for all the genes were found to be less than 1, which again showed purifying selection is prevailing over all the seven genes. Ka/Ks values ranged from 0–1 which also indicated synonymous selection is more prevalent over these seven genes. Tajima's D values were also found to be negative for (*egl*, *fli*C, *gdh*A, *leu*S and concatenated sequences) and non-significant positive values were observed for the genes *adk* and *gyr*B. This finally inferred that the genes of worldwide RSSC are under purifying selection.

## Genetic diversity among different agro-climatic zone of West Bengal

Analysis of diversity of RSSC among the five agro-climatic zones of West Bengal was done using DnaSP v6.12 and Arlequin v3.5 to evaluate fixation index (Fst), nucleotide diversity (π) and haplotype diversity (Hd) (Table 5). A total of 16 haplotypes of RSSC were observed over

**Table 5. Standard genetic diversity indices for RSSC populations of five agroclimatic zones.**

| Population Id | Description of Zone | N[b] | Fst[c] | Π[d] | S[e] | Hd[f] | H[g] | Tajima's D | N_T[h] | N_R[i] |
|---|---|---|---|---|---|---|---|---|---|---|
| ACZ2[a] | Testa Tarai alluvial zone | 4 | 0.20 | 0.001 | 6 | 0.83 | 3 | 1.16 | 6 | 0 |
| ACZ3 | Gangetic Alluvial zone | 14 | 0.25 | 0.0005 | 12 | 0.27 | 3 | -2.17* | 10 | 2 |
| ACZ4 | Old Alluvial zone | 9 | 0.16 | 0.0011 | 13 | 0.83 | 6 | -0.559 (0.31) | 10 | 3 |
| ACZ5 | Coastal Saline zone | 3 | 0.22 | 0.0009 | 5 | 0.67 | 2 | 0.0 (0.799) | 5 | 0 |
| ACZ6 | Red and lateritic zone | 6 | 0.06 | 0.0021 | 18 | 1.00 | 6 | -0.27 (0.425) | 13 | 5 |

[a]Agro Climatic Zone

[b] Number of isolates

[c]Fixation index

[d] Nucleotide diversity

[e]Number of segregating sites

[f] Haplotype diversity

[g] Number of haplotypes

[h] Number of Transitions

[i] Number of Transversions

*(p<0.05)

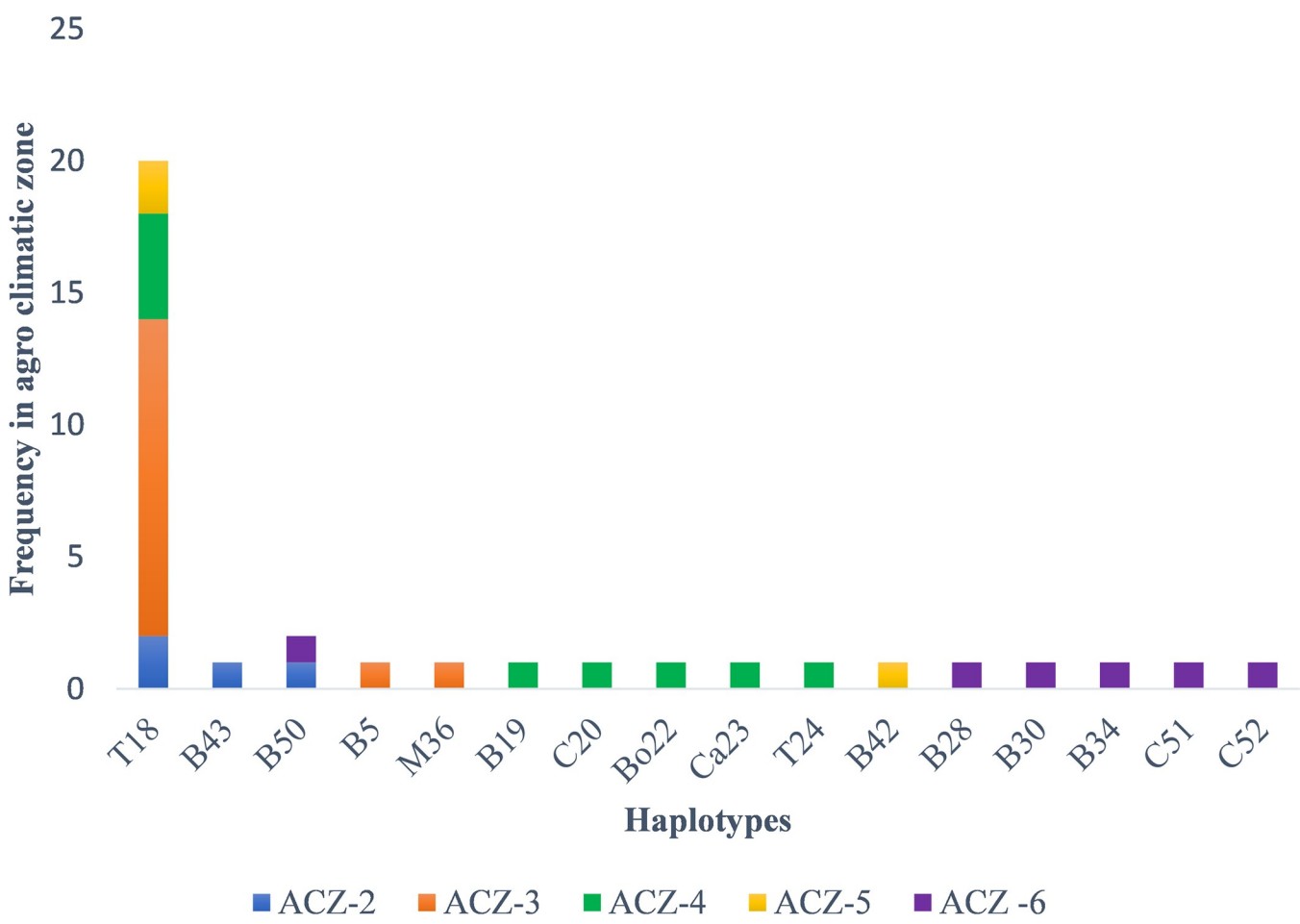

**Fig 8. Haplotype frequency among the populations of five agro climatic zones, based on MLSA (the name of the haplotypes are one of the representative isolates within the haplotypes).**

the 5 agro-climatic zones with a mean haplotype diversity of 0.69. The maximum number of haplotypes were found in ACZ-6 (H = 6) and ACZ-4 (H = 6), however, the haplotype diversity was maximum for the population of ACZ-6 (Hd = 1) which indicates that all six isolates (n = 6) of ACZ-6 are different haplotypes. Hence, the population of ACZ- 6 is highly diverse in West Bengal. Moderately high haplotype diversity was observed for the population of ACZ 4 (Hd = 0.83) and ACZ 2 (Hd = 0.83). The agro climatic zones ACZ-2 and ACZ-3 comprised of 3 haplotypes each. The least number of haplotypes were prevalent in ACZ-5. Isolate numbers varied from 1 to 12 for a haplotype. Except ACZ-6, the haplotype T18 was found to be prevalent and also in multiple frequency in all other agroclimatic zones. The haplotype B50 appeared in both ACZ-2 and ACZ-6. The rest 14 haplotypes were unique for different agroclimatic zones, represented by single isolate (Fig 8).

Also, on the basis of nucleotide diversity, transitions and transversions, maximum diversity was observed for RSSC population of ACZ-6 [nucleotide diversity ($\pi$) = 0.21%, number of transitions ($N_T$) = 13 and number of transversions ($N_R$) = 5] followed by ACZ-4 ($\pi$ = 0.11%, $N_T$ = 10 and $N_R$ = 3) (Table 5). Moderate level of diversity was observed for the population in ACZ-5 [nucleotide diversity ($\pi$) = 0.09%, number of transitions ($N_T$) = 5 and number of transversions ($N_R$) = 0] and ACZ 3 ($\pi$ = 0.05%, $N_T$ = 10 and $N_R$ = 2). The least diversified population of RSSC was ACZ-2 ($\pi$ = 0.1%, $N_T$ = 13 and $N_R$ = 5). Tajima's D values for ACZ-3, ACZ-4

**Table 6. AMOVA for concatenated sequences of seven genes of RSSC populations from five agro-climatic zones of West Bengal.**

| Source of variation | Df | Sum of squares | Variance components | Variation | Fixation index | p-value |
|---|---|---|---|---|---|---|
| Among populations | 4 | 18.329 | 0.42 Va | 18.90 | 0.189 | 0.00782+-0.00242 |
| Within Populations | 31 | 55.698 | 1.79 Vb | 81.10 | | |
| Total | 35 | 74.028 | 2.21544 | | | |

and ACZ-6 were negative whereas, non-significant positive Tajima's D values were observed for RSSC in ACZ-2 and ACZ-5.

Analysis of Molecular Variance (AMOVA) conducted by using Arlequin v3.5 revealed that only 18.9% of the genetic variation is present among agroclimatic zones, whereas, higher genetic variation of 81.1% is observed within the agroclimatic zones (Table 6). The pairwise Fst value for genetic distance was observed to be maximum and significant between the populations ACZ-3 and ACZ-6 (pairwise Fst = 0.47206*) followed by the Fst value between populationsACZ-6 and ACZ-4 (pairwise Fst = 0.26125*) (Table 7; Fig 9).

All the RSSC belonging to Phylotype I in West Bengal (C1 = 36) and worldwide collection (C2 = 76) were collectively analysed (C3 n = 112) to understand the diversity among all the four phylotypes using DnaSP v6.12 (Table 8). The highest nucleotide diversity was obtained in case of phylotype IV (π = 2.06%) followed by phylotype IIA (π = 1.15%), phylotype III (π = 1.08%), phylotype IIB (π = 0.95%) and the minimum in case of the phylotype I (π = 0.5%). Again, Tajima's D values within the range of 0–1 indicate occurrence of purifying selection over each gene of all the four phylotypes.

The highest nucleotide diversity was observed for the gene *adk* in phylotype I (π = 0.92%), *fli*C gene in Phylotype IIA (π = 2.2%), *adk* gene in phylotype IIB (π = 9.7%) and *egl* gene in phylotype III (π = 1.69%), whereas, the genes *fli*C, *pps*A and *gdh*A possessed highest nucleotide diversity (π = 2.46% each) in phylotype IV.

Analysis of molecular variance (AMOVA) conducted by Arlequin v3.1 revealed that Phylotype-I and Phylotype IIB are genetically most dissimilar from each other (Fst: 0.87464**), while Phylotype IIA and IIB are genetically closer (Fst: 0.43599**) (Table 9). Again, variation among phylotypes (51.41%) is higher than within phylotypes (13.08%). Further, population specific Fst values computed by AMOVA revealed that maximum sharing of genetic materials is taking place in Phylotype IV with the least value of fixation index (Fst = 0.786). The least sharing of genetic material was observed for Phylotype I with highest fixation index (Fst = 0.80) [30] (Table 10).

### Estimation of intra and inter phylotypic recombinations

Recombination test of the concatenated sequences consisting seven genes of the 36 RSSC West Bengal isolates was performed with SplitsTree 4 program. The pairwise homoplasy test

**Table 7. Pairwise Fst (below diagonal), population specific Fst (bold values at diagonal) and p values (above diagonal) between RSSC populations of five agro climatic zones of West Bengal.**

| | ACZ 2* | ACZ 3 | ACZ 4 | ACZ 5 | ACZ 6 |
|---|---|---|---|---|---|
| ACZ 2 | **0.20256** | 0.11035+-0.0107 | 0.25391+-0.0167 | 0.99902+-0.0002 | 0.34473+-0.0132 |
| ACZ 3 | 0.20318 | **0.24819** | 0.23438+-0.0137 | 0.43555+-0.0140 | 0.00000+-0.0000 |
| ACZ 4 | 0.06677 | 0.02484 | **0.16305** | 0.88086+-0.0081 | 0.00684+-0.0027 |
| ACZ 5 | -0.31677 | -0.00623 | -0.0891 | **0.22336** | 0.32715+-0.0131 |
| ACZ 6 | 0.03471 | 0.47206* | 0.26125* | 0.0655 | **0.06359** |

* ACZ denotes Agro climatic zones

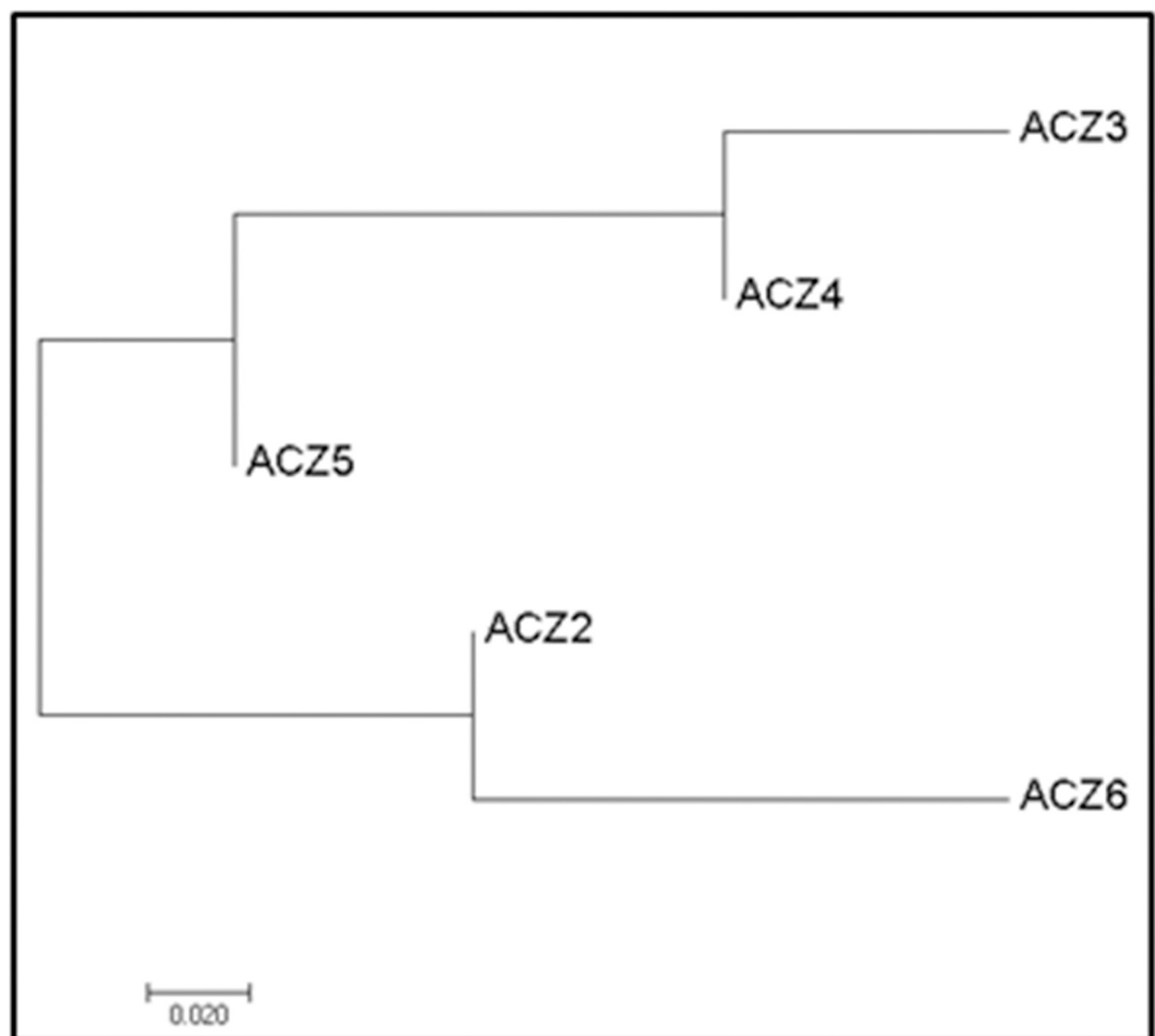

**Fig 9. Clustering of agro climatic zones on basis of pairwise genetic distance (Fst) Genetic diversity among phylotypes of RSSC.**

revealed significant recombination ($\theta = 0.47^*$) among the isolates of *R. solanacearum* in West Bengal (Table 11). The splits tree was constructed using 1000 boot strap analysis. The 36 isolates of West Bengal formed reticulated network which indicates occurrence of recombination among the strains of five agro climatic zones, belonging to Phylotype I. The Neighbour Net tree revealed a total of 29 splits with 3564 characters (Fig 10). The estimated invariable sites were 0.00285 with delta score of 0.027.

Further, considering the concatenated sequences of these seven genes, we have investigated the rate of recombination for all the phylotypes of worldwide collection (C3 = 112) of *R. solanacearum*. Reconstruction of evolutionary relationship using Neighbour Net analysis of Splits Tree 4 showed prominent reticulations, which in turn indicated presence of recombinations within genes of the phylotypes of RSSC (Fig 11).

**Table 8. Genetic diversity of the of the worldwide RSSC isolates (C3 = 112) at phylotypic level.**

| Phylotypes | Gene | TS[a] | NS[b] | S[c] | P[d] | K[e] | Eta[f] | π[g] | H[h] | Hd[i] |
|---|---|---|---|---|---|---|---|---|---|---|
| I | *adk* | 381 | 56 | 18 | 4.72 | 3.49 | 20 | 0.009 | 15 | 0.731 |
| | *egl* | 654 | 56 | 19 | 2.91 | 2.65 | 19 | 0.004 | 17 | 0.77 |
| | *flic* | 312 | 56 | 5 | 1.60 | 0.34 | 5 | 0.0011 | 6 | 0.204 |
| | *gdh*A | 534 | 56 | 6 | 1.12 | 0.65 | 6 | 0.0012 | 6 | 0.457 |
| | *gyr*B | 354 | 56 | 23 | 6.50 | 6.91 | 25 | 0.0019 | 16 | 0.772 |
| | *leu*S | 699 | 56 | 5 | 0.72 | 0.73 | 5 | 0.001 | 5 | 0.525 |
| | *pps*A | 630 | 55 | 11 | 1.74 | 1.58 | 11 | 0.0025 | 9 | 0.632 |
| | Concatenated | 3564 | 56 | 76 | 2.59 | 14.777 | 80 | 0.00504 | 32 | 0.85 |
| IIA | *adk* | 381 | 24 | 19 | 4.99 | 3.23 | 21 | 0.0085 | 11 | 0.88 |
| | *egl* | 654 | 24 | 43 | 6.57 | 10.66 | 43 | 0.0163 | 17 | 0.953 |
| | *flic* | 312 | 24 | 18 | 5.77 | 6.8 | 19 | 0.022 | 17 | 0.949 |
| | *gdh*A | 534 | 24 | 17 | 3.18 | 4.63 | 17 | 0.0086 | 8 | 0.87 |
| | *gyr*B | 354 | 24 | 19 | 5.37 | 4.91 | 20 | 0.014 | 13 | 0.917 |
| | *leu*S | 699 | 24 | 16 | 2.29 | 3.49 | 16 | 0.005 | 11 | 0.87 |
| | *pps*A | 630 | 20 | 16 | 2.54 | 4.28 | 16 | 0.007 | 10 | 0.889 |
| | Concatenated | 3564 | 24 | 132 | 4.49 | 33.761 | 136 | 0.01151 | 24 | 1 |
| IIB | *adk* | 381 | 15 | 13 | 3.41 | 3.71 | 13 | 0.097 | 11 | 0.952 |
| | *egl* | 654 | 15 | 28 | 4.28 | 10 | 28 | 0.0153 | 6 | 0.82 |
| | *flic* | 312 | 15 | 12 | 3.85 | 3.96 | 12 | 0.0127 | 8 | 0.90 |
| | *gdh*A | 534 | 15 | 2 | 0.37 | 0.952 | 2 | 0.00178 | 2 | 0.48 |
| | *gyr*B | 354 | 15 | 9 | 2.54 | 3.49 | 9 | 0.0088 | 9 | 0.886 |
| | *leu*S | 699 | 15 | 14 | 2.00 | 5.66 | 14 | 0.0081 | 6 | 0.82 |
| | *pps*A | 630 | 14 | 6 | 0.95 | 2.35 | 6 | 0.0037 | 4 | 0.70 |
| | Concatenated | 3564 | 15 | 78 | 2.66 | 27.771 | 78 | 0.00947 | 14 | 0.99 |
| III | *adk* | 381 | 13 | 13 | 3.41 | 3.31 | 13 | 0.0086 | 8 | 0.91 |
| | *egl* | 654 | 13 | 32 | 3.41 | 11.03 | 34 | 0.0169 | 12 | 0.9 |
| | *flic* | 312 | 13 | 5 | 4.28 | 1.744 | 5 | 0.0056 | 6 | 0.82 |
| | *gdh*A | 534 | 13 | 24 | 3.85 | 6.346 | 25 | 0.0012 | 12 | 0.987 |
| | *gyr*B | 354 | 13 | 13 | 0.37 | 3.86 | 15 | 0.011 | 11 | 0.962 |
| | *leu*S | 699 | 13 | 25 | 2.54 | 5.54 | 25 | 0.0079 | 11 | 0.974 |
| | *pps*A | 630 | 12 | 34 | 5.39 | 9.88 | 34 | 0.0157 | 8 | 0.972 |
| | Concatenated | 3564 | 13 | 112 | 3.82 | 31.821 | 117 | 0.01085 | 13 | 1 |
| IV | *adk* | 381 | 4 | 9 | 3.41 | 5 | 9 | 0.01312 | 4 | 1 |
| | *egl* | 4 | 4 | 28 | 4.89 | 14.17 | 27 | 0.0216 | 3 | 0.833 |
| | *flic* | 312 | 4 | 15 | 1.60 | 7.667 | 15 | 0.0246 | 3 | 0.833 |
| | *gdh*A | 534 | 4 | 26 | 4.49 | 13.17 | 26 | 0.0246 | 3 | 0.833 |
| | *gyr*B | 354 | 4 | 13 | 3.67 | 7.83 | 13 | 0.0221 | 3 | 0.833 |
| | *leu*S | 699 | 4 | 24 | 3.58 | 12.83 | 24 | 0.0184 | 3 | 0.833 |
| | *pps*A | 630 | 3 | 22 | 3.49 | 15 | 23 | 0.0246 | 3 | 1 |
| | Concatenated | 3564 | 4 | 115 | 3.92 | 60.667 | 115 | 0.02068 | 4 | 1 |

[a] total number of sites

[b] number of sequences

[c] number of polymorphic sites

[d] Percentage of polymorphism

[e] Average number of nucleotide differences

[f] Nucleotide diversity

[g] Number of mutations

[h] Number of Haplotypes.

[i] Haplotype diversity

*(p<0.05)

**Table 9. Pairwise Fst of phylotypes based on genetic distance.**

| Population ID | Phylotype—I | Phylotype—IIA | Phylotype—IIB | Phylotype—III | Phylotype—IV |
|---|---|---|---|---|---|
| Phylotype–I | 0 | | | | |
| Phylotype–IIA | 0.85925** | 0 | | | |
| Phylotype–IIB | 0.87464** | 0.43599** | 0 | | |
| Phylotype–III | 0.73927** | 0.73546** | 0.76286** | 0 | |
| Phylotype–IV | 0.82639** | 0.75024** | 0.76905** | 0.69944** | 0 |

The phylotype I was observed to further diverge into three groups, *viz*., Phylotype I.1, Phylotpe I.2 (containing isolates from West Bengal) and Phylotype I.3. The Phylotype IIA also diverged into two clusters, namely IIA.1 and IIA.2. This shows that the two phylotypes are still expanding. The assumption was further confirmed by conducting pairwise homoplasy test (θ) using the Splits tree 4 programme [25]. The test (θ) revealed occurrence of significant recombination in all phylotypes except for phylotype IV (Table 12).

Analysis using RDP4 programme [24] revealed eleven significant recombination events occurring between the phylotypes (Fig 12) (S3 Table). A recombination event was considered significant if it was detected with two or more methods of recombination point identification. The gene *pps*A was not considered for recombination analysis as many reference sequences of *R. solanacearum* for the gene *pps*A are lacking. The other six genes *viz*., *adk*, *egl*, *fli*C, *gyr*B, *gdh*A and *leu*S were considered for analysis among which all others except *leu*S exhibited evidence of recombination. The Fig 12 explains the recombination events taking place within and between the phylotypes. A total of 105 strains of *R. solanacearum* were identified as recombinants out of total 112 (C3) worldwide strains. A maximum of four recombination events (RE—4,8,10 and11) were observed in the 33 recombinants of Phylotype III, followed by three different recombination events in Phylotype I, IIA, and IV each. Only one recombination event (RE– 9) was observed for phylotype IIB. Hence, the most recombinogenic among the monophyletic clusters of RSSC is Phylotype III. Among the five phylotypes, maximum number of recombinant strains (n = 65) were observed in Phylotype I. Most frequent recombination event was RE-11, which took place between Phylotype IIA with Unknown parent and Phylotype IIB with Unknown parent (S3 Table). Most of the donor parents were observed from the Phylotype- I and Phylotype- II.

On the basis of recombination to mutation ratio (ρ/θ), the seven genes were categorised in two different groups. In the whole set (all phylotypes combined), the rate of recombination was less than mutation (ρ/θ<1) for the genes *egl*, *gdh*A and *leu*S. Contrarily, the rate of recombination was more than the rate of mutation (ρ/θ>1) for the genes *gyr*B, *fli*C, *pps*A and *adk*. Among the house keeping genes *gyr*B and *pps*A possessed comparatively higher recombination to mutation rate and among the genes of megaplasmid, high recombination to mutation rate was observed for *fli*C gene (Table 12).

In order to consider both the indels and substitutions in phylogenetic analysis for obtaining maximum evolutionary information from the sequences generated in the current experiment,

**Table 10. Phylotype specific fixation index (Fst) computed by AMOVA with concatenated sequences.**

| Population | Fst |
|---|---|
| Phylotype I | 0.80102 |
| Phylotype IIA | 0.79274 |
| Phylotype IIB | 0.79592 |
| Phylotype III | 0.79343 |
| Phylotype IV | 0.78586 |

**Table 11. Pair wise homoplasy index test (Ø) value obtained from Splits Tree 4 programme.**

| Population ID | Ø value | p value |
|---|---|---|
| Phylotype-I | 0.42* | 0.001 |
| Phylotype -IIA | 0.25* | 5.42E-6 |
| Phylotype -IIB | 0.29* | 1.92E-7 |
| Phylotype -III | 0.36* | 1.46E-4 |
| Phylotype -IV | 0.1 | 0.6342 |
| C1(n = 36) | 0.47* | 0.0079 |

alignment of the concatenated sequences of the seven genes for all the phylotypes of *R. solanacearum* were utilised and R package SIDIER has been used to interpret both indels and substitution matrix and evolutionary relationship was expressed as percolation networks.

Percolation network depicted in Fig 13 has identified 91 haplotypes from 112 concatenated sequences of *R. solanacearum* from worldwide. Through analysis of combined distance of indels and substitutions, these 91 haplotypes were diversified into 3 groups with percolation threshold of 0.58. Fifty haplotypes of RSSC were separated in the first group (represented in red) which consisted of 36 haplotypes of Phylotype I, 13 haplotypes of Phylotype III and a single haplotype of Phylotype IV. The second group (represented in yellow), consisting 38 haplotypes, was constituted by 24 haplotypes from Phylotype IIA and 14 haplotypes from phylotype IIB. The last group (represented in green) consisted of 3 haplotypes from Phylotype IV.

## Discussions

### Status of bacterial wilt incidence under West Bengal condition

Survey was conducted over five different agro climatic zones of West Bengal during January, 2017 to December, 2018 and maximum incidence of bacterial wilt was observed in the red and

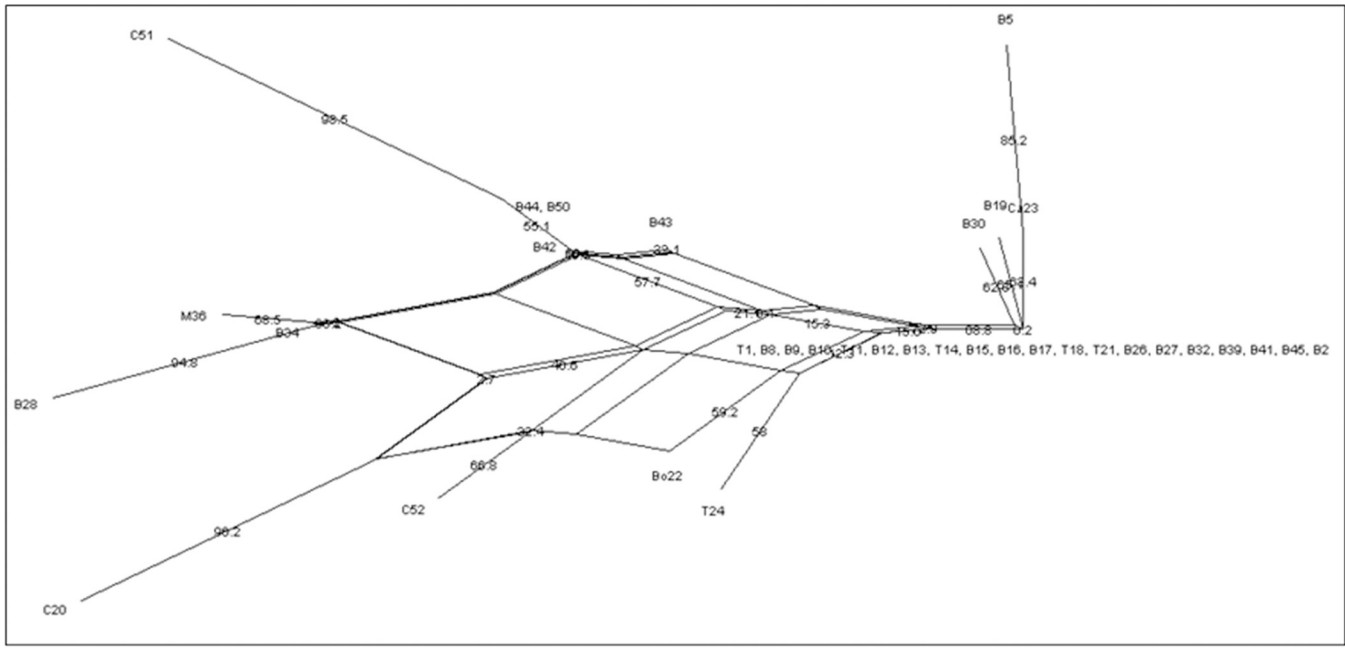

**Fig 10. Neighbour net tree constructed using concatenated sequences of the seven genes (*adk*, *egl*, *fli*C, *gyr*B, *gdh*A, *leu*S and *pps*A) for 36 isolates of RSSC from West Bengal (C1) with the SplitsTree 4 programme with scale bar of 0.04 using 1000 bootstrap.**

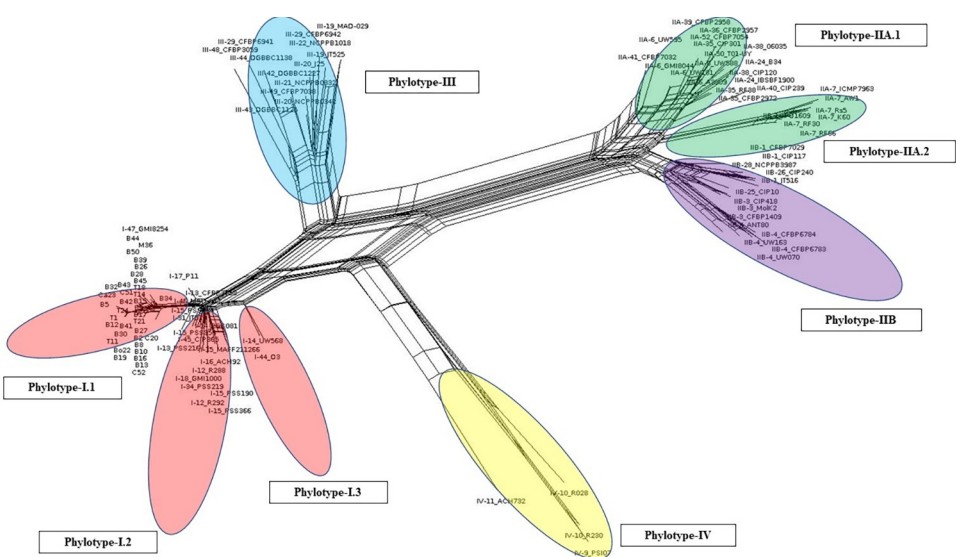

**Fig 11. Neighbour net tree constructed using concatenated sequences of the seven gene (*adk, egl, fliC, gyrB, gdhA, leuS and ppsA*) for 36 isolates of RSSC from West Bengal (C1) and reference sequences from 76 strains of worldwide collection (C2) for illustration of evolutionary relationships among them with the SplitsTree 4 programme.**

lateritic zone (ACZ-6) (PDI = 42.4%) followed by old alluvial zone (PDI = 40%) and minimum in Coastal saline zone (ACZ-5) (PDI = 26.9%). The ACZ-6 is characterised by high acidic pH 4.5–5.5, low phosphorus content in soil and hot and sub-humid climatic conditions [31]. Again, increase in bacterial wilt severity in soil pH of range 4.5 to 5.5 has been recorded [32]. Due to these and several other factors explained in the later sections, high incidence of bacterial wilt was observed in ACZ-6.

The first report of bottlegourd (*Lagenaria siceraria*) as a host of *R. solanacearum* was done from China [33]. In our present investigation, we have reported the bacterial wilt of bottlegourd from India and bacterial wilt of marigold (*Tagetes erecta*) from West Bengal for the first time. The presence of bacterial wilt of marigold was earlier reported from Andaman Islands of India [34]. In addition to this, bacterial wilt was observed in brinjal, chilli, capsicum and tomato in the present investigation.

**Table 12. Estimates of mutation (θ) and recombination (ρ) rates using LDhat programme of RDP4.**

| Gene | (θ) | | | | | | (ρ) | | | | | | (ρ/ θ) | | | | | |
|---|---|---|---|---|---|---|---|---|---|---|---|---|---|---|---|---|---|---|
| | I* | IIA | IIB | III | IV | Whole set | I | IIA | IIB | III | IV | Whole set | I | IIA | IIB | III | IV | Whole set |
| *adk* | 4.92 | 5.80 | 4.44 | 5.01 | 4.91 | 5.26 | 5.08 | 0.62 | 20.71 | 1.293 | 10.66 | 12.96 | 1.032 | 0.106 | 4.666 | 0.257 | 2.172 | 2.46 |
| *egl* | 5.62 | 12.7 | 12.2 | 9.93 | 18.6 | 33.44 | 22.71 | 1.75 | 0.081 | 17.5 | 0.007 | 2.07 | 4.042 | 0.137 | 0.006 | 1.762 | 0.0003 | 0.06 |
| *flic* | 2.19 | 5.03 | 4.63 | 2.19 | 10 | 7.48 | 112.2 | 9.09 | 5.075 | 573.8 | 0.003 | 28.48 | 51.23 | 1.808 | 1.097 | 262 | 0.0050 | 3.81 |
| *gdhA* | 2.63 | 6.55 | 2 | 7.61 | 17.3 | 7.85 | 111.7 | 1.09 | 424.7 | 16.62 | 0.012 | 49.80 | 42.55 | 0.1662 | 212.3 | 2.183 | 0.0007 | 0.22 |
| *gyrB* | 6.33 | 5.8 | 3.31 | 3.76 | 8.6 | 18.27 | 35.46 | 14.4 | 559.1 | 826.5 | 0.103 | 4.07 | 5.604 | 2.48 | 168.8 | 220.0 | 0.0119 | 6.34 |
| *leuS* | 2.19 | 5.462 | 6.132 | 8.536 | 16 | 17.97 | 858.7 | 0.07 | 0.102 | 0.953 | 0.039 | 9.51 | 392.1 | 0.0135 | 0.017 | 0.111 | 0.002 | 0.53 |
| *ppsA* | 3.88 | 5.65 | 3.27 | 13.11 | | 21.28 | 1.25 | 3.05 | 0.25 | 3.84 | | 80.15 | 0.29 | 0.53 | 0.07 | 0.29 | | 3.76 |
| Concatenated | 20.02 | 34.28 | 26.41 | 34.48 | 62.73 | | 15.84 | 10.25 | 14.67 | 148.66 | 7.33 | | 0.79 | 0.29 | 0.55 | 4.31 | 0.11 | |

* I, IIA, IIB, III and IV are phylotypes of *R. solanacearum*

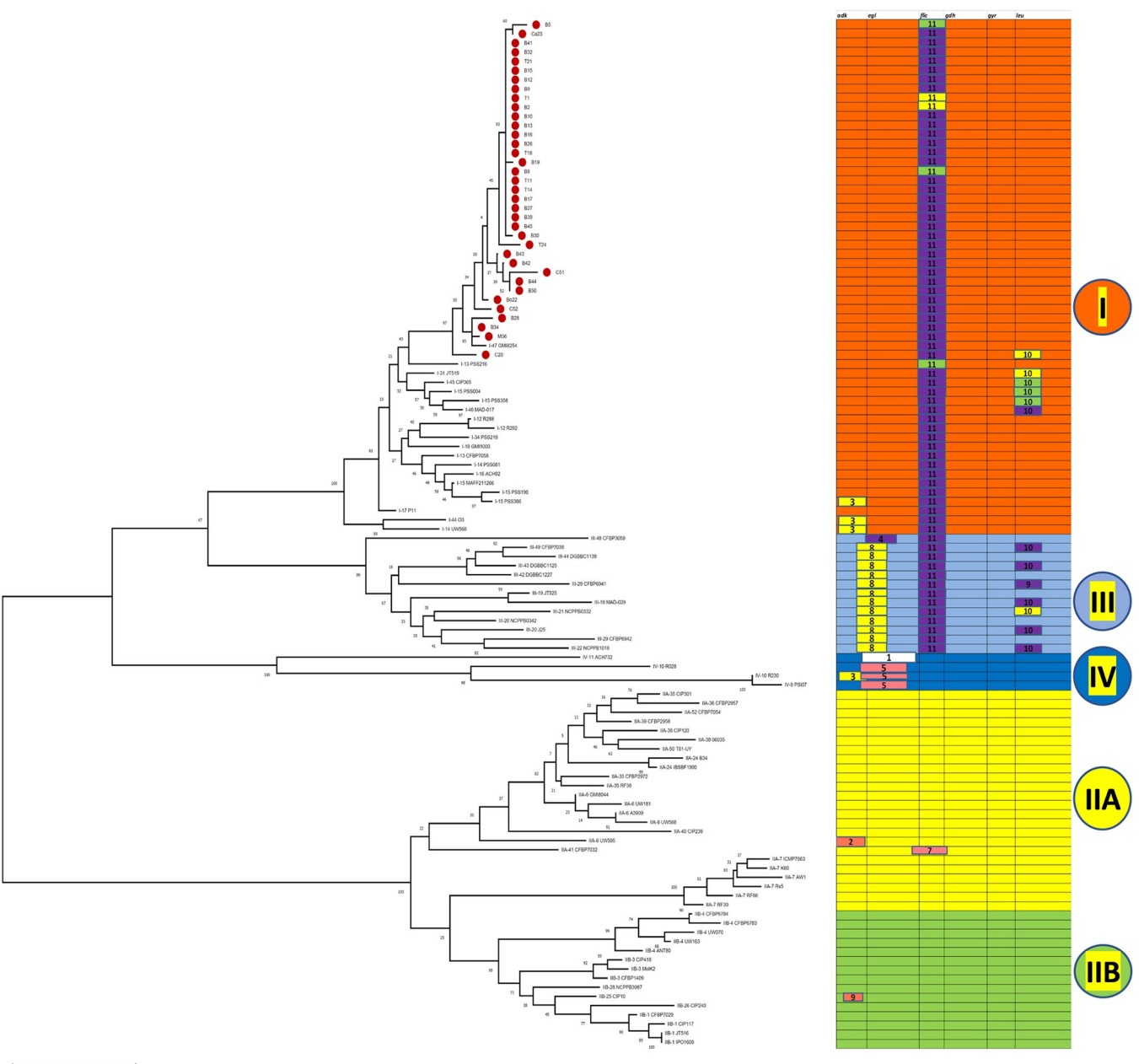

**Fig 12. Maximum-likelihood tree of 112 *Ralstonia solanacearum* strains constructed with MEGA 7 using concatenated sequences of seven loci (*adk*, *egl*, *fli*C, *gyr*B, *gdh*A, *leu*S and *pps*A), representing recombination events identified with RDP 4 [26] coloured according to the phylogenetic position of the donor (minor parent).**

## Prevalence of race and biovar across five agro-climatic zones of West Bengal

Investigation was undertaken to unravel the genetic diversity of *Ralstonia solanacearum* species complex (RSSC) prevailing under different agro-ecological conditions of West Bengal. All the 36 isolates of RSSC collected from 17 districts of West Bengal and inciting wilt on brinjal, chilli, capsicum, tomato, bottlegourd and marigold (C1) were categorised as Race 1. Hence,

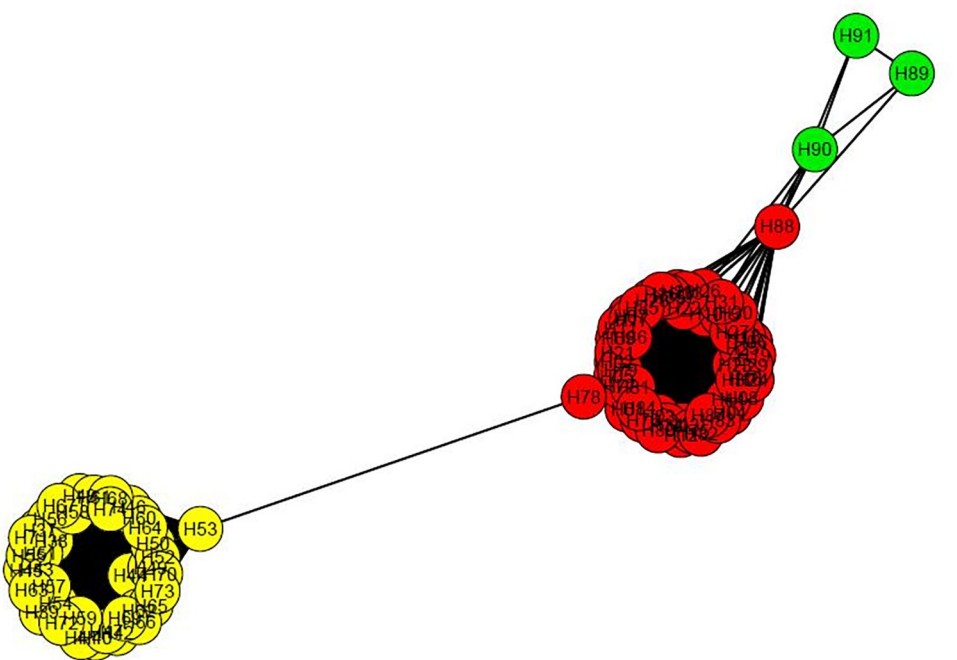

**Fig 13. Percolation network generated by SIDIER module diversified 91 haplotypes from 112 sequences into 3 groups with percolation threshold of 0.58.**

prevalence of race 1 of *R. solanacearum* was confirmed across West Bengal. Earlier [35], reported the presence of race 1 of *R. solanacearum* in West Bengal, but only from 3 districts *viz.*, Purulia, Burdwan and Birbhum.

Among biovars, in addition to biovar 3, the prevalence of biovar 6 and 3b was observed across the five agro-ecological regions of West Bengal. The most predominant biovar among the isolates collected is biovar 3b (50%) followed by biovar 6 and biovar 3 (25% each). Earlier, the biovar 6 was reported from China [36], which is unable to utilise dulcitol. The presence of biovar 3b was first observed from Kerala, India, inciting wilt on tomato and ginger [37]. These isolates were similarly unable to use dulcitol and lactose, as found in our research work. Hence, for the first time, we report the prevalence of biovar 6 from India and biovar 3b from the state of West Bengal, India. The distribution of these three biovars of *R. solanacearum* was found to be uneven across the five-vegetable growing agro climatic zones of West Bengal. In addition to these three biovars, presence of biovar 2 has also been reported from Coochbehar district of West Bengal [38].

### Phylotype and sequevar status

Sequencing of the 16s rDNA and phylogenetic tree analysis categorised all the collected isolates of West Bengal into Division 1, which according to the new hierarchical system is Phylotype I [5]. Again, multiplex phylotype PCR also confirmed all collected isolates to be Phylotype I,

which is composed of Asiatic strains possessing the widest host range among all the four phylotypes. Evolutionary studies conducted by Wicker and colleagues stated that the Phylotype I has its origin with maximum probability in North Asia followed by primary migration to East Asia and East Africa [29]. This may be the major reason behind high prevalence of Phylotype I in India. Hence, identification of new hosts (bottlegourd and marigold) in West Bengal, India, is the evidence of the continuous diversification and spreading of Phylotype I of *R. solanacearum*.

The identification of sequevar is done on the basis of *egl* and *hrp*B genes. The reasons behind the wide adoption of endoglucanase gene (*egl*) for sequevar identification are: the gene is one of the major factors of virulence, *R. solanacearum* possess only a single copy of the gene in its genome, the gene is conserved and ubiquitous in all the strains with sufficient level of variation enabling separation of the *R. solanacearum* strains into sequevars [5, 39]. On the other hand, *hrp*B gene is responsible for the stimulation of hypersensitive reactions in resistant hosts and expression of disease symptoms in susceptible host plants. PCR-RFLP studies with this gene have been shown to evolve parallelly with 16s rRNA and tRNA gene and possess variability that is helpful for discriminating strains of *R. solanacearum* [17]. Hence the *egl* and *hrp*B genes are widely used to characterise *R. solanacaerum* into sequevars. Till date seven sequevars are reported to be randomly distributed over different states of India. Sequevar I-45 and I-30 were reported from Himachal Pradesh, IIB-1 from Uttar Pradesh, Madhya Pradesh, Meghalaya and West Bengal, I-48 from Maharashtra, I-47 and I-17 from Karnataka, I-17, I-14 and I-48 from Goa, I-17 and I-47 from Kerala and I-14, I-47 and I-48 from Andaman & Nicobar Islands [40].

The sequevar distribution was also not restricted to agroclimatic zones of West Bengal and was found to be randomly distributed. The prevalence of the two sequevars I-47 and I-48 was reported for the first time from the five agroclimatic zones of West Bengal, in addition to the previously described potato infecting sequevar IIB-1from ACZ-2 [38]. The sequevar I-48 was found to be predominant (75%) over the five-vegetable growing agroclimatic zones of West Bengal and the sequevar I-47 was distributed with remaining 25 per cent frequency (Fig 14). Such irregular distribution of sequevars has also been reported from various parts of the world like Mayotte island of Indian Ocean [8].

The sequevar I-48 has been reported to have a wide host range with Mulberry in China [41], potato and tomato in Myanmar [42] and brinjal and chilli in some parts of India [40]. In addition to this, our investigation revealed that this sequevar also infects bottle gourd (*Lagenaria siceraria*) (*R. solanacearum* isolate Bo22) and Bell pepper (*Capsicum annuum*) (*R. solanacearum* isolate Ca23). The sequevar I-47 have been reported to be pathogenic on tomato in Indonesia and brinjal and chilli in India [40]. I-47 could also infect Marigold (*Tagetes erecta*) in West Bengal, India, which was observed for the first time in our current study.

## Interpretation of genetic diversity in RSSC of West Bengal and world

Multi locus sequence analysis of the concatenated sequences of seven genes of 112 isolates (C1 and C2 combined) revealed all the genes of RSSC to be under purifying selection. In the West Bengal collection (C1), high nucleotide diversity was observed for the genes *gyrB*, followed by *adk*, *egl*, *pps*A, *fli*C, *gdh*A and the least in *leu*S. On the other hand, among the worldwide RSSC collection, high nucleotide diversity was exhibited by the genes *egl* and *pps*A followed by similar nucleotide diversity in remaining five genes. Hence, contrasting diversity was found for the genes of RSSC between West Bengal, India (C1) and worldwide (C2). C1 is comprised of only Phylotype-I which is distributed across five agro-climatic zones of West Bengal. However, worldwide collection C2 is comprised of all the phylotypes from various geographical location

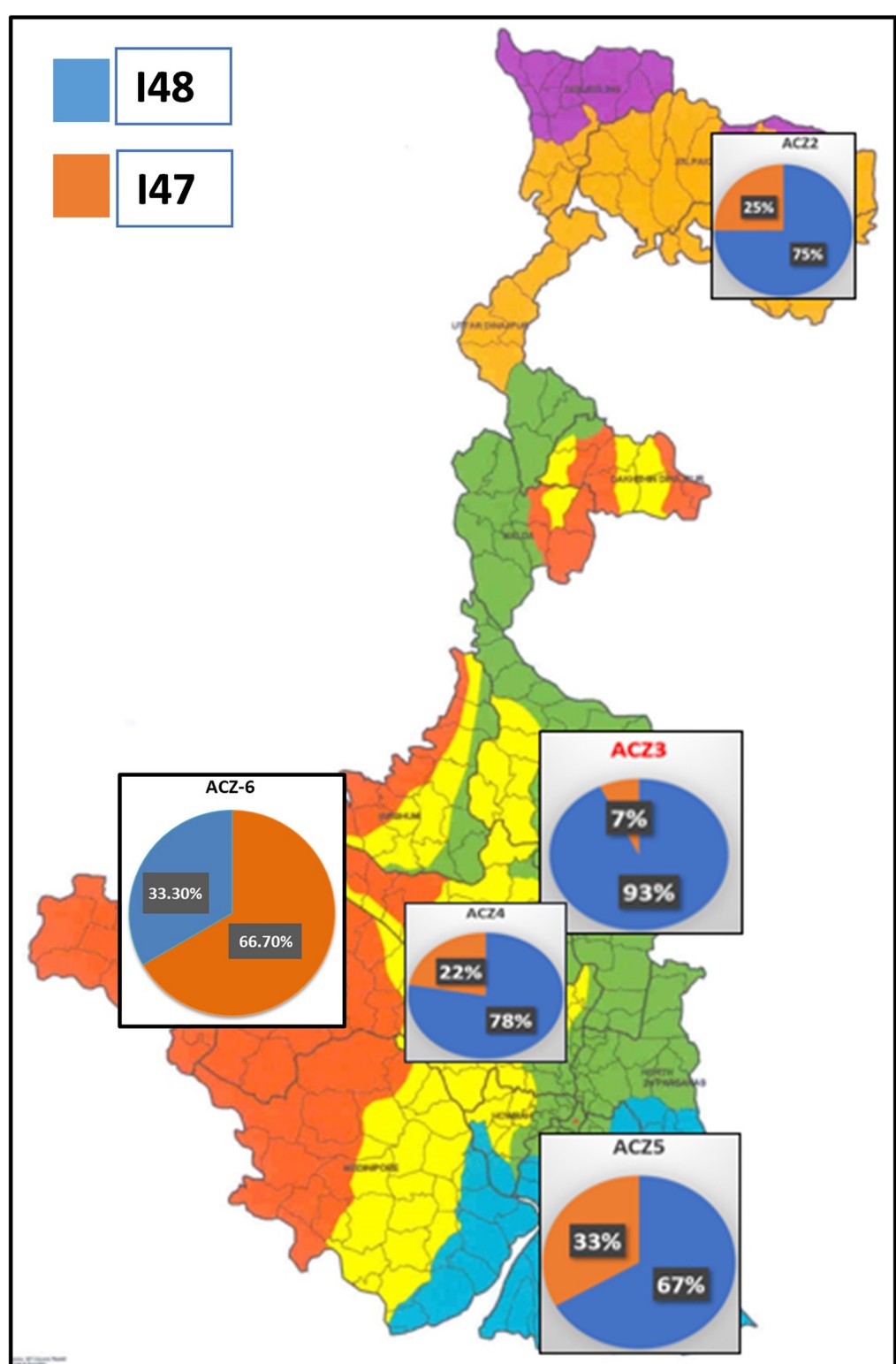

**Fig 14. Map of West Bengal showing relative abundance of sequevars in different agroclimatic zones of West Bengal.** Reprinted from [Redefined six Agro-climatic sub-regions of West Bengal,India] under a CC BY license, with permission from [Pranab 2018], original copyright [2001].

of the world. This may be the reason for contrasting diversity exhibited among C1 and C2 collections.

Under C1 collection, the value for synonymous substitutions were obtained for all the genes except *leuS* where as values for non-synonymous substitution were obtained for few of the considered genes (Table 4). This indicates lack of equal evolutionary pressure across various genes considered in the current experiment. Evolutionary divergence was not large enough to produce non-synonymous substitutions in all the considered genes, although synonymous substitutions were observed for all of them except *leu*S. Hence, purifying or synonymous selection was found to act over the genes of RSSC isolates collected from West Bengal. In comparison to C1, the evolutionary pressure acting on C2 was large enough to produce both synonymous and non synonymous substitution values. But less than 1 values for $K_a/K_s$ and $d_N/d_S$ also indicates purifying or synonymous selection acting over worldwide collection of RSSC (C2).

## Genetic diversity and gene flow among agro climatic zones of West Bengal

Maximum number of population diversity parameters *viz*., haplotypes, haplotype diversity, nucleotide diversity and nucleotide substitution revealed that the population of RSSC in the Red and Lateritic Zone of West Bengal (ACZ-6) is the most diverse, followed by Old Alluvial Zone (ACZ-4). Analysis of Molecular Variance (AMOVA) also revealed that the individuals are most freely breeding among themselves in the ACZ-6 with the least fixation index (Fst = 0.06) followed by ACZ-4 (Fst = 0.16). This illustrates the high genetic diversity in the population of ACZ-6 and ACZ-4. This Red and Lateritic Zone of West Bengal (ACZ 6) shares border with the Jharkhand state, where solanaceous vegetable crops are widely cultivated. Regular exchange of seed materials of solanaceous vegetables is practised between the state of Jharkhand and several districts of West Bengal in ACZ-6 such as Purulia, Paschim Bardhaman, Bankura, Birbhum and Jhargram. The Purulia district accounts for the major part of ACZ-6, is also the extension of the Chota Nagpur Plateau Region (Manbhum-Singhbhum Division) that has its major part in the state of Jharkhand. Bankura district serves as the connecting link between Chota Nagpur Plateau region and the Gangetic plains of West Bengal. As compared to other agro climatic zones of West Bengal, ACZ-6 has more area of acidic soils (pH 4.18 to 6.6) rich in iron and aluminium oxides [43], because the laterization has occurred from granite-gneiss parent rocks of the extended Chota Nagpur Plateau region. Hot and subhumid climate with maximum temperature of 45˚C and minimum of 9˚C and hyperthermic soils (mean annual soil temperature is 22˚C or higher) is the characteristics of this Red and Lateritic agro climatic zone (ACZ-6) [32]. These soils are deficient in available phosphorous content. It was reported that the bacterium *R. solanacearum* thrives better in acidic soil with pH ranging from 4.5–5.5 [31] and the soil $P_2O_5$ content has been reported to be negatively correlated (r = -0.53) with the vascular bacterial wilt incidence. All the soil and climatic factors of ACZ-6 well supports the occurrence of high bacterial wilt incidence thereby higher genetic diversity of RSSC. In contrast, the population of RSSC in the ACZ-5 (Fst = 0.22) and ACZ-2 (Fst = 0.20) were comparatively more fixed and thus less diverse. The maximum fixation index was observed for the population of New Gangetic Alluvial zone–ACZ-3 (Fst = 0.25) and hence is genetically least diverse population (Table 5).

High within population and low among population variation infers possibility of gene flow and connectivity among the populations of RSSC of five agro climatic zones of West Bengal (Table 6). Significantly high pairwise Fst value (genetic distance) suggested restricted gene flow between ACZ-6 (Red and Lateritic zone)–ACZ-3 (New Gangetic Alluvial zone) and between ACZ-6 –ACZ-4 (Old Gangetic Alluvial zone). The reason behind this restricted gene

flow which may confer population diversification between these zones may be attributed to different edapho-climatic conditions prevailing in the various agro climatic zones. The pair wise homoplasy test showed the presence of recombination among the strains of RSSC of the five agro climatic zones of West Bengal. Reticulated network observed in the Neighbour Net tree with 29 splits also strengthens the evidences of gene flow or genetic exchange among these regions. The RE 11 identified by the RDP 4 program shows that the 36 isolates of West Bengal are recombinants produced by genetic exchange between donor Phylotype IIA, IIB and unknown recipient parents. Again, the RE 10 shows that the isolate of chilli (C 20) collected from West Bengal is a recombinant strain. This strain was generated as a result of genetic exchange between donor Phylotype IIA, IIB and recipient C1 (West Bengal isolates), belonging to Phylotype-I. In three REs (2, 5 and 7), these 36 isolates of West Bengal were involved as donor parent in recombination with the strains of other phylotypes *viz*., Phylotype IIA and IIB (S3 Table). In the RE—3, 8 and 10, the isolates of *R. solanacearum* of West Bengal received genetic materials from the isolates of Phylotype IIA and IIB. Hence, the isolates of West Bengal belonging to Phylotype I are involved in exchange of genetic material with isolates of Phylotype II. The genetic exchange between the Phylotype I and Phylotype II is possibly due to the occurrence of Phylotype-II inciting brown rot of potato in West Bengal [38]. Studies showed similar reticulated network formation in neighbour net tree of Phylotype I of *R. solanacearum* collected from Goa, Orissa and Maharashtra states of India [44]. Hence, Indian strains of Phylotype I are engaged in recombination with strains belonging to other phylotypes.

## Diversity among the phylotypes of RSSC worldwide

Our findings of genetic diversity among phylotypes revealed highest nucleotide diversity in Phylotype IV followed by Phylotype IIA, Phylotype III, Phylotype IIB and minimum in Phylotype I. The Maximum genetic diversity in Phylotype IV is also exhibited by its least fixation index (Fst = 0.786) as computed by AMOVA in this study. Conversely, maximum fixation index of Phylotype I (Fst = 0.801) of RSSC signifies the lowest genetic diversity among all the phylotypes of *R. solanacearum* (Table 10). The study conducted by Yahiaoui and his colleagues with the RSSC strains of South West Indian Ocean Island and worldwide collection also revealed the similar trend of maximum nucleotide diversity for Phylotype IV followed by Phylotype II, Phylotype III and Phylotype I [45].

The evolutionary trees were constructed with MEGA 7 separately for each seven genes (*adk*, *egl*, *fli*C, *gyr*B, *gdh*A, *leu*S and *pps*A) and with the concatenated sequences. Surprisingly, the trees showed incongruence with each other on their topography. Neighbour net network construction using Splitstree4 [25] revealed presence of many parallelograms, which shows recombination events across and within the phylotypes. Pairwise homoplasy index (PHI) values obtained were significant (p<0.005) for all the phylotypes confirming prevalence of recombination, which may be responsible for their genetic evolution. Further, Splits tree analysis showed that the Phylotype I has diverged into 3 groups (I.1, I.2 and I.3). This divergence shows that the Phylotype I is still under expansion. The expanding nature of Phylotype I shown in earlier studies may be due to high recombination, primary migration to Eastern Africa and eastern Asian regions and secondary migrations to America, Europe, Asia and Western African countries [27]. Similarly, Neighbour net tree indicated divergence of Phylotype IIA into two groups, but was not found in Phylotype IIB. Population specific fixation index (Table 10) shows that the population of Phylotype IIB is more fixed as compared to that of Phylotype IIA. This explains the reason behind expanding nature of Phylotype IIA as compared to Phylotype IIB. The expanding nature of Phylotype II has also been confirmed in earlier studies [46].

Investigation of recombination by RDP4 programme finally confirmed 11 significant recombination events occurring across the phylotypes. The highest recombination to mutation ratio (ρ/θ) explains the reason behind maximum number of varied recombination events in Phylotype III. Moderate rate of recombination was observed in Phylotype I (Table 12) as high number of major parents (159 strains) and minor parents (65 strains) involved in recombination. Moderate recombination rate of Phylotype I is also supported by the fact that they have wider and increasing host range [47]. Comparatively lower recombination rate was observed in Phylotype IV and Phylotype-IIA. Among genes, mutation rate was higher for the genes *egl*, *gdh*A and *leu*S. High recombination rate was observed for the genes *gyr*B, *fli*C, *pps*A and *adk*.

The amount of evolutionary information is usually limited due to low divergence between closely related organisms in phylogenetic studies. Phylogenetic analysis is usually based on nucleotide substitutions [48]. While insertions and deletions (indels) provide additional information for the reconstruction of sequence evolution. Consideration of indels is becoming more and more important in DNA barcoding and phylogenetic analysis which is ignored in contemporary analysis of the same. Percolation theory has been utilized in the phylogeographic analysis of animals [49, 50], pollination niches and floral divergence in various plants and genetic structure of endemic plants [51]. However, application of this theory in agriculture has been very limited. Das and colleagues has reviewed the worldwide structure of various species of *Stemphylium* through the application of percolation theory and identified relationship among various species [52].

In our study, clustering with percolation network grouped the five phylotypes of RSSC into three groups. It was observed that the strains of Phylotype I and Phylotype III grouped into a single cluster with only a single out-numbered isolate of Phylotype IV (ACH732). Comparatively lower pairwise genetic distance between these Phylotype I and Phylotype-III (Fst = 0.73927\*\*) explains the phenomenon of their clustering into the same group. Again, the least pairwise genetic distance (Fst = 0.43599\*\*) supports the clustering of Phylotype IIA and Phylotype IIB into the same cluster.

## Conclusion

With the finding *Lagenaria siceraria* and *Tagetes erecta* as new hosts and existence of more biovars and sequevars earlier not identified implies continuous diversification and spreading of Phylotype-I in West Bengal, India. In West Bengal, gene flow is occurring among the agro climatic zones and the isolates of *R. solanacearum* are involved in recombination. Diversity of *R. solanacearum* population in West Bengal and worldwide is contrasting and the genes are under purifying selection.

## Supporting information

**S1 Table. Information of primers and PCR conditions used for amplification of genes for the conformation of *R. solanacearum* and MLSA study.**
(DOCX)

**S2 Table. Accession numbers of R. solanacearum partial gene sequences retrieved from NCBI.**
(DOCX)

**S3 Table. Identification of recombination events among various phylotypes of *R. solanacearum*.**
(CSV)

**S1 Fig. Amplification of 36 isolates of *R. solanacearum* producing species-specific amplicon of approximately 288 bp fragment with OLI1-Y2 primer pairs in 1.5% agarose gel. Lane L: 100 bp ladder.**
(DOCX)

**S2 Fig. Amplification of 36 isolates of *R. solanacearum* producing amplicon of approximately 1500 bp of the 16S rDNA regions with 27F-1525R primer pairs in 0.8% agarose gel. Lane L: 100 bp ladder.**
(DOCX)

**S3 Fig. Gel electrophoresis of phylotype specific multiplex PCR for six isolates of *R. solanacearum* collected from various ACZ of West Bengal. The isolates produced amplicon of 144 bp specific to Phylotype I and 288 bp specific to *R. solanacearum*.**
(DOCX)

**S4 Fig. Partial amplification of endoglucanase gene (*egl*) with EglF-EglR primer pairs producing approximately 850 bp fragment. Lane L: 10?0 bp ladder.**
(DOCX)

**S5 Fig. Partial amplification of *hrp*B gene with 36 *R. solanacearum* isolates with *RShrpBf—RShrpBr* primer pairs producing approximately 1417 bp fragment. Lane L: 100 bp ladder.**
(DOCX)

## Acknowledgments

We are thankful to the Plant Bacteriology Laboratory, Department of Plant Pathology of Bidhan Chandra Krishi Viswavidyalaya for providing laboratory and instrumental facilities.

## Author Contributions

**Conceptualization:** Subrata Dutta, Ashis Roy Barman.

**Data curation:** Ankit Kumar Ghorai.

**Formal analysis:** Ankit Kumar Ghorai, Ashis Roy Barman.

**Funding acquisition:** Ashis Roy Barman.

**Investigation:** Ankit Kumar Ghorai, Subrata Dutta.

**Methodology:** Ankit Kumar Ghorai, Subrata Dutta, Ashis Roy Barman.

**Project administration:** Ashis Roy Barman.

**Resources:** Ashis Roy Barman.

**Software:** Subrata Dutta, Ashis Roy Barman.

**Supervision:** Subrata Dutta.

**Validation:** Subrata Dutta, Ashis Roy Barman.

**Visualization:** Subrata Dutta.

**Writing – original draft:** Ankit Kumar Ghorai.

**Writing – review & editing:** Subrata Dutta, Ashis Roy Barman.

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
