## [Decision Letter · Decision Letter 0]

6 Jul 2022

PONE-D-22-05127Genetic diversity of Ralstonia solanacearum causing vascular bacterial wilt under different agro-climatic regions of West Bengal, IndiaPLOS ONE

Dear Dr. Roy Barman,

Thank you for submitting your manuscript to PLOS ONE. After careful consideration, we feel that it has merit but does not fully meet PLOS ONE’s publication criteria as it currently stands. Therefore, we invite you to submit a revised version of the manuscript that addresses the points raised during the review process.

Dear authors,

In line with the reviewer comment I too find the reported study by Ghorai et. al 2022 very interesting.  The research work reported here contains extensive field work, laboratory work and bioinformatics analysis to reach the conclusions. I must appreciate the authors for this wonderful research.

However, I have some minor concerns, which are listed below.

Line no 22- diseases can be corrected to disease

Line no 23- (Smith 1896) Yabuuchi et al. 1995 citation may not be required in the abstract

Line no 89- “reacting positive” change wording

Line no – “32 ±1°C or 27 ±1°C day/night” are the authors want to mean day temperature 32 ±1°C and 27 ±1°C as night temperature? Clarify and if necessary modify.

Line no – 164“Purification of extracted DNA was made by centrifugation at 15493g for 5 min” I think better not to refer the process as purification since it is just a simple centrifugation step.

Line no – 164  “16S-23SITS”, after 23s give a space

Line no – 737- “This Red and Lateritic Zone of West  Bengal shares the major border with solanaceous vegetable growing neighbouring state  Jharkhand, India.” Rewrite the sentence.

Line no 317: Pathogenecity test- Among 36 isolates, results of all the isolates are not mentioned properly. Please rectify.

Line no 462: “No values were obtained for non-synonymous selection” is it appropriate?

Please check.

Best Regards

Niraj Agarwala

We look forward to receiving your revised manuscript.

Kind regards,

Niraj Agarwala, Ph.D.

Academic Editor

PLOS ONE

https://journals.plos.org/plosone/s/file?id=ba62/PLOSOne_formatting_sample_title_authors_affiliations.pdf".

“We are thankful to the Science and Engineering Research Board (SERB), Department of Science and Technology, Government of India for providing financial support in the form of ECR Award.”

“P.I. of the Project: Ashis Roy Barman

Grant Number: File No. ECR/2017/000547, dated 2610.2017

Funding Authority: Science and Engineering Research Board (SERB), Department of Science and Technology, Government of India

URL: http://www.serb.gov.in/home.php

3. We note that [Figure 14] in your submission contain [map/satellite] images which may be copyrighted. All PLOS content is published under the Creative Commons Attribution License (CC BY 4.0), which means that the manuscript, images, and Supporting Information files will be freely available online, and any third party is permitted to access, download, copy, distribute, and use these materials in any way, even commercially, with proper attribution. For these reasons, we cannot publish previously copyrighted maps or satellite images created using proprietary data, such as Google software (Google Maps, Street View, and Earth). For more information, see our copyright guidelines: http://journals.plos.org/plosone/s/licenses-and-copyright.

a. You may seek permission from the original copyright holder of Figure 14 to publish the content specifically under the CC BY 4.0 license. 

Natural Earth (public domain): http://www.naturalearthdata.com/.

Reviewers' comments:

Reviewer's Responses to Questions

**Comments to the Author**

1. Is the manuscript technically sound, and do the data support the conclusions?

Reviewer #1: Yes

2. Has the statistical analysis been performed appropriately and rigorously? 

Reviewer #1: Yes

3. Have the authors made all data underlying the findings in their manuscript fully available?

Reviewer #1: Yes

4. Is the manuscript presented in an intelligible fashion and written in standard English?

Reviewer #1: Yes

5. Review Comments to the Author

Reviewer #1: Bacterial wilt in plants caused by Ralstonia solanacearum is prevalent in India. Understanding the pathogenicity potential of natural isolates in different hosts and grouping them by molecular analysis is an important study, which has been carried out by the authors. I found the methodology followed is rigorous and very interesting findings in the manuscript. I recommend its pub;ication.

6. PLOS authors have the option to publish the peer review history of their article (what does this mean?). If published, this will include your full peer review and any attached files.

Reviewer #1: No

---

## [Author Response · Author response to Decision Letter 0]

20 Aug 2022

All the specific comments are being addressed in the Response to Reviewers.docx file uploaded. About the funding information, the Funding Statement is sufficient and all the funding information has been removed from the acknowledgement section.

We are thankful to the Editor and Reviewers for considering this manuscript for reviewing and further looking forward for the final consideration of the manuscript.

---

## [Editor Report · Decision Letter 1]

6 Sep 2022

Genetic diversity of Ralstonia solanacearum causing vascular bacterial wilt under different agro-climatic regions of West Bengal, India

PONE-D-22-05127R1

Dear Dr. Roy Barman,

We’re pleased to inform you that your manuscript has been judged scientifically suitable for publication and will be formally accepted for publication once it meets all outstanding technical requirements.

Kind regards,

Niraj Agarwala, Ph.D.

Academic Editor

PLOS ONE
---

## [Editor Report · Acceptance letter]

12 Sep 2022

PONE-D-22-05127R1 

Genetic diversity of *Ralstonia solanacearum* causing vascular bacterial wilt under different agro-climatic regions of West Bengal, India 

Dear Dr. Roy Barman:

I'm pleased to inform you that your manuscript has been deemed suitable for publication in PLOS ONE. Congratulations! Your manuscript is now with our production department. 

Kind regards, 

on behalf of

Dr. Niraj Agarwala 

Academic Editor

PLOS ONE